# Causes and Effects of Unanticipated Numerical Deviations in Neural Network Inference Frameworks

**Alexander Schlögl**         **Nora Hofer**         **Rainer Böhme**

Department of Computer Science
Universität Innsbruck, 6020 Innsbruck, Austria

`{alexander.schloegl | nora.hofer | rainer.boehme}` @ `uibk.ac.at`

## Abstract

Hardware-specific optimizations in machine learning (ML) frameworks can cause numerical deviations of inference results. Quite surprisingly, despite using a fixed trained model and fixed input data, inference results are not consistent across platforms, and sometimes not even deterministic on the same platform. We study the causes of these numerical deviations for convolutional neural networks (CNN) on realistic end-to-end inference pipelines and in isolated experiments. Results from 75 distinct platforms suggest that the main causes of deviations on CPUs are differences in SIMD use, and the selection of convolution algorithms at runtime on GPUs. We link the causes and propagation effects to properties of the ML model and evaluate potential mitigations. We make our research code publicly available.

## 1 Introduction

The "reproducibility crisis" machine learning is facing, and potentially fueling [18], has drawn attention to efforts that improve reproducibility. They include checklists [27] and guidelines [5; 33], benchmarks designed with reproducibility in mind [10; 4; 22], reproducibility contests [32], and repositories like PapersWithCode [30]. However, we find that even with a fully defined environment and without stochastic processes, runtime optimizations of ML frameworks can cause deviations in inference results. These are outside of researchers' control, and cannot be fully avoided at present.

Uncontrolled numerical deviations are detrimental to many aspects of ML. If deviations occur systematically, key assumptions in federated learning [7; 25], heterogeneous ML [36], and proof-of-learning [11] may not hold. The fact that platforms leave fingerprints in the inference results opens new possibilities for forensics [34]. Finally, numerical deviations have implications on ML security: specifically crafted "boundary samples" may trigger label flips depending on the hardware [35].

To study the causes and effects of these deviations, we instrument the popular TensorFlow framework (version 2.5.0) on various layers. All our experiments are containerized and automatically deployed and executed on 75 distinct hardware configurations hosted on the Google Cloud Platform (GCP), Amazon Web Services (AWS), and on our premises. (Details of our setup are in Section B of the supplementary material.) For a fixed trained model and input, the 75 platforms produced up to 26 different softmax vectors in the last layer. As label flips remain rare, the existence of deviations is not apparent in typical performance metrics, such as test set accuracy.

We make three contributions:

1. We offer the so-far most comprehensive evaluation of causes and effects of (known) numerical deviations in CNN inference, spanning a wide range of heterogeneous platforms.
2. We are the first to associate causes of deviations with properties under control of the ML engineer, such as floating-point precision, layer type, or activation function.

37th Conference on Neural Information Processing Systems (NeurIPS 2023).

3. We make the code of our infrastructure[1] and experiments[2] publicly available, allowing follow-up researchers to measure deviations between runs and platforms and inspect them layer by layer. The set of supported platforms can be adjusted with limited effort.

The body of this paper is structured as follows. Section 2 provides background, establishes terminology, and reviews related work. Section 3 presents the experimental results. Starting from end-to-end inference pipelines, we walk through individual causes for CPUs (Section 3.1), GPUs (Section 3.2), link them to properties of the ML model (Section 3.3), and study potential mitigations (Section 3.4), always supported with experiments. Section 4 discusses the findings and Section 5 concludes.

## 2 Background

The main causes for deviations in inference results are different aggregation orders at finite precision, and the use of different approximate convolution algorithms.

**Aggregation order**    The order in which arithmetic operations are executed can affect the result for limited precision. Consider an example in a toy decimal floating point representation with *only one* significant digit. In this representation, integers $0 \leq |x| \leq 10$ can be represented exactly. For values $10 < |y| \leq 100$, $y$ must be rounded to the next multiple of 10, and the least significant digit is lost. We will denote rounding to the nearest representable value with $[x]$.

$$(a + b) + c = [[7 + 4] - 5] = [1\text{E}1 - 5] = 5 \tag{1}$$
$$a + (b + c) = [7 + [4 - 5]] = [7 - 1] = 6 \tag{2}$$

The above example shows how the aggregation order can change the result. In Equation (1), $[7+4] = [11]$ is rounded to $1\text{E}1 = 10$. This effect is known as *swamping* [13]. Hence, optimizations that change the aggregation order depending on the hardware can cause deviations between platforms.

**Convolution algorithms**    Recall the formula for 2D convolution,

$$R_{i,j} = \sum_{n=0}^{h} \sum_{m=0}^{w} I_{i-m,j-n} F_{m,n}, \tag{3}$$

for a 2-dimensional filter of size $h \times w$ with input $I$ and filter $F$. To reduce the size of the convolution result, sometimes a stride is applied to the convolution, which is multiplied with input indices $i, j$.

The naive implementation of convolution is a nested loop expressing the nested sums. This approach is often inefficient as spatial proximity in the input does not imply locality in memory, in particular for higher dimensions. Modern hardware uses multiple layers of caching, which depend on locality and coalesced access for maximum performance. Due to the prevalence of convolution, especially in ML, many optimized implementations are available [2; 3].

All optimizations reduce convolution to generalized matrix multiplication (GEMM) [6]. GEMM convolution, the simplest optimization, extracts the relevant parts of the array into a Toeplitz matrix, replicates the filter as needed, and multiplies the two matrices. GEMM convolution exists in precomputed, implicit, and explicit variants [3]. By contrast, *Winograd* short convolution [9] transforms both inputs and filters to achieve a lower number of multiplications. There exist fused and non-fused variants [40]. As the transformations used for Winograd convolution are inherently lossy, and rounding occurs at multiple stages, results may deviate. Convolutions using fast Fourier transformation (FFT) exploit the fact that spatial convolution is equivalent to point-wise multiplication in the frequency domain. The transformation to and from the frequency domain is inherently lossy for finite numerical precision [24] and introduces deviations from other methods. In addition, modern GPUs include special TensorCores to compute convolution directly [28].

A more detailed description of convolution approaches can be found in Section C of the supplementary material. Performance characteristics of the different approaches are excellently explained in Blahut et al. [6]. Specific performance characteristics for NVidia GPUs are discussed in [29].

---

[1]https://github.com/uibk-iNNference/iNNfrastructure
[2]https://github.com/uibk-iNNference/unaNNticipated

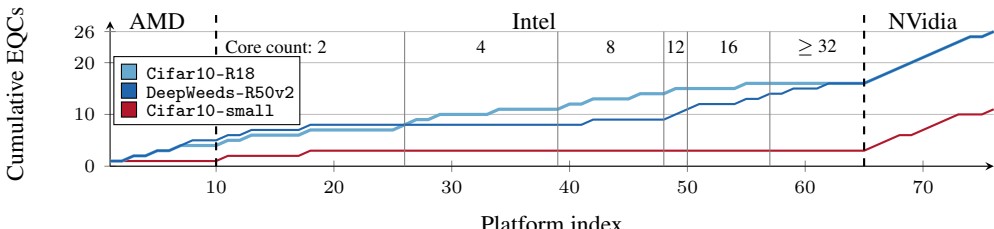

Figure 1: Main result: cumulative number of different softmax vectors in the last layer over all platforms. Platforms are sorted by vendor, architecture, and core count. Core count for Intel CPUs.

**Conventions** We use *architecture* as shorthand for microarchitecture. Architecture, core count, and memory size together form a *platform*. We call inference *deterministic* if the same input, the same trained model, and the same platform always produce the same output. We say it is *consistent* if the same input and model always produce the same output across all platforms. We denote our models as `Dataset-Architecture`, e.g, `Cifar10-R18` is a ResNet18 [15] trained on Cifar-10 [21].

To measure the effect of numerical deviations, we use the concept of *equivalence classes* (EQCs): platforms that produce identical outputs form an EQC. The number of EQCs is a measure of diversity. Ideally, all platforms fall into a single EQC, meaning that inference is deterministic and consistent (as expected in theory). To quantify the magnitude of deviations, we use the *remaining precision*. This metric counts the number of identical mantissa bits before the first deviation between two or more (intermediate) results. For single precision floating-point numbers, it is in the range $[0, 23]$. The metric generalizes to tensors by taking the minimum of the remaining precision of all elements.

**Related work** Much work has been done to improve reproducibility in ML [33; 5; 10; 4; 22]. These works make important contributions to the organizational aspects of reproducibility, but do not address the problem of numerical deviations. The literature has examined the influence of variance in algorithm and implementation on model *training*. Pham et al. [31] find that variance can lead to significant differences in model performance and training time. Zhuang et al. [41] extend the experiments and find differences in training performance across different GPUs and TPUs. This end-to-end approach is beneficial for the community and informs practitioners of the potential impact of variances in algorithm and implementation on the final model. Our work differentiates itself by focusing on *inference* and by drilling down to the root causes of the observed deviations.

Deviations in ML inference have been reported in the signal processing community by Schlögl et al. [34], however with a focus on forensics. Our earlier work offers existential evidence from a few CPU platforms, but does not investigate causes or mitigation strategies. GPUs are also not considered.

The computer arithmetic community is well aware of the non-associativity of floating point computations [26; 19], but aims to increase precision and efficiency rather than enforcing associativity.

## 3 Influences on model outputs

In total, 64 of our 75 platforms were CPU-based. Depending on the model, the softmax vector of the last layer produced between 3 and 16 different EQCs. This means all tested models failed to produce consistent outputs across the CPU platforms. We did not observe indeterminism on CPUs. The remaining 11 of our 75 platforms supported inference on GPUs. The softmax outputs produced between 8 and 10 different EQCs. Every EQC had cardinality one, meaning the GPU could be uniquely identified by the numerical deviations in the softmax vector. 39 out of 99 inference outputs on GPUs were indeterministic. Figure 1 shows the cumulative number of EQCs over all platforms in this scenario (see Tables SUP-1 and SUP-2 for all hardware, model, and input details).

While the main results above are end-to-end measurements for realistic inference pipelines, it is instructive to study individual causes with specifically designed experiment on a reduced set of platforms. All following results are supported with isolated experiments. We discuss the influences affecting model outputs on CPUs and GPUs in turn.

Table 1: Different CPUs produce deviations based on their SIMD capabilities.

| | EQC | 0 | 1 | 2 | 3 | 4 | 5 | 6 | 7 | 8 | 9 | 10 | 11 | 12 | 13 | 14 |
|---|---|---|---|---|---|---|---|---|---|---|---|---|---|---|---|---|
| | | | | | | | | | | | | Flag cluster | | | | |
| Intel Sandy Bridge | 1 | | | | × | | | | | | | | | | | × |
| Intel Ivy Bridge | 1 | | | | × | | | | | | × | × | | | | × |
| Intel Haswell | 2 | ×∗ | | | × | | | | | | × | × | | × | | × |
| Intel Broadwell | 2 | × | | | × | | | × | | | × | × | × | × | | × |
| Intel Skylake | 3 | × | ×† | | | × | × | | × | | × | × | × | × | × | × |
| Intel Cascade Lake | 3 | × | × | | | × | × | | × | ×● | × | × | × | × | | |
| Intel Ice Lake | 4 | × | × | ×‡ | | × | × | × | × | × | × | × | × | × | | |
| AMD Rome | 5 | × | | | ×§ | | × | × | × | | | × | | | | |
| AMD Milan | 5 | × | | | × | | × | × | × | | × | × | | × | | |

∗ contains 256-bit SIMD flags; † contains some 512-bit SIMD flags; ‡ contains more 512-bit SIMD flags; § contains `sse4a` (128-bit SIMD) and `misalignsse` flags; ● contains the `avx512vnni` flag

## 3.1 Influences when inferring on CPUs

Model outputs computed on CPUs may deviate because of differences in data and task parallelism. Both affect the aggregation order of convolutions.

**Data parallelism** Modern architectures feature a variety of SIMD instructions, which affect both the floating-point accuracy and the aggregation order. While CPUs traditionally compute floating-point operations at very high precision in the FPU, the introduction of SIMD instructions, such as `SSE` and `AVX` on x86, enables data parallelism at the cost of reduced precision. Newer CPUs have larger SIMD registers, which allow more flexibility when adjusting the SIMD width (i. e., data parallelism) and desired precision. Some CPUs support fused multiply-and-add in SIMD [16; 1].

To measure the effect of data parallelism on CPU, we perform inference with the `Cifar10-R18` model on all dual-core x86 systems available on GCP, and collect all CPUID flags indicating hardware support of SIMD instructions. Since the number of flags (169) exceeds the number of platforms, we cluster flags that always co-occur. Table 1 shows the relation between EQCs and flag clusters.

Observe that a subset of the flag clusters perfectly aligns with the EQCs. Flag clusters 0–3 are all related to the support for different SIMD capabilities. Flag cluster 0 contains the `avx2` flag, denoting 256-bit SIMD support. Flag cluster 1 contains the `avx512f` flag, indicating an SIMD width of 512 bit. Flag cluster 2 contains four SIMD-related flags: `avx512vbmi` (vector bit manipulation instructions), `avx512ifma` (fused multiply-add for integers), `vpclmulqdq` (carry-less multiplication), and `avx512vpopcntdq` (population count). While the co-occurrence prevents us from attributing the deviation to a single flag, we consider this evidence to show that some SIMD-related feature is responsible for the EQC split. Flag cluster 3 contains the `sse4a` flag not present in the other CPUs, as well as the `misalignsse` flag to indicate support for misaligned memory access when using legacy SIMD instructions. For this EQC, however, the effect of different SIMD support may be superseded by general architectural differences of AMD processors. Interestingly, no new EQC emerges based on the support for `avx512vnni` vector neural-network instructions, indicating that the framework does not use this hardware feature yet.

**Task parallelism** Divide-and-conquer algorithms can increase performance and reduce memory overhead by distributing work to multiple cores [38]. The use of task parallelism involves additional aggregation steps, the order of which can vary depending on the implementation, e. g., atomic versus sum–reduce. As work gets distributed on more cores, the individual workload decreases. When the workload per core becomes too small, data parallelism cannot be used anymore. Figure 2 visualizes this effect. When increasing core count from 2 to 4, the additional 512-bit SIMD capabilities of the Intel Ice Lake platform become unusable, producing the same outputs as Intel Sky-/Cascade Lake. Going from 8 to 16 cores makes the per-core workload too small for 256-bit SIMD instructions, resulting in a single EQC for all Intel platforms newer than Ivy/Sandy Bridge. We did not observe additional EQC collapses for core counts larger than 16. Both the Intel Sandy/Ivy Bridge

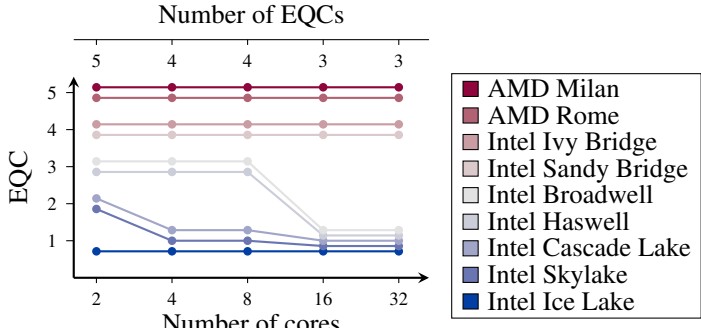

Figure 2: Deviations disappear as the number of CPU cores increases.

and the AMD Rome/Milan architectures have sufficiently different SIMD capabilities (cf. Table 1) to produce deviations even for larger core counts.

## 3.2 Influences when inferring on GPUs

The choice of convolution algorithm is a cause of deviations on GPUs. Due to different performance characteristics, there is no universally best convolution algorithm [6]. Modern GPUs implement a large number of convolution algorithms, and select the fastest algorithm for the convolution parameters (number of filters, stride, etc.) at runtime based on microbenchmarks.[3] Therefore, the final algorithm choice may not only vary with the GPU and model architecture, but also with the remaining hardware and even uncontrollable conditions, like bus contention and parallel load. As each convolution is benchmarked separately, different layers of a model can use different algorithms.

**Deviations between GPUs**  We perform inference with the `Cifar10-R18` model on all GPUs available on GCP and locally. Using the TensorFlow profiler, we log all function calls and extract the convolution algorithms. Internally, the TensorFlow profiler uses the NVidia profiler to record calls to the CUDA API and some CUDA-internal function calls. A list of all observed convolution algorithms can be found in Table SUP-4 in the supplementary material.

For the GPUs with deterministic outputs, we can explain the deviations by the use of different approximate convolution algorithms. Figure 3a shows exemplary traces from the GTX 1650 and RTX 2070 GPUs on our local machines. While 16 different convolution functions were called in our experiments, we group them by their approach for ease of reading. The lines in Figure 3a plot the selected algorithm for each convolutional layer of the model. In this case, both GPUs choose the same algorithm for each of the first 15 convolutions. Convolution 16 is computed with different algorithms, but they happen to produce identical results. This is because the only difference between explicit and implicit GEMM is the fact that explicit stores the Toeplitz matrix in memory, whereas implicit GEMM computes it on the fly. The deviations are caused by the final three convolutions, where the RTX 2070 continues to use implicit GEMM while the GTX 1650 switches back to Winograd. The separation of EQCs is indicated by the dashed vertical line, which we verify by comparing the intermediate results. We confirm from all other traces that this pattern is typical for deterministic deviations between GPUs.

**Deviations between inferences on the same GPU**  Microbenchmarks are run just before the first inference. The framework uses random data to fill a buffer of the problem dimensions and measures the execution time of all supported convolution algorithms. Since these measurements take up valuable execution time, each candidate algorithm is timed only once. This makes the choice of the "winner" susceptible to runtime conditions, e. g., bus contention and memory latency.

To measure variations between inferences, we perform repeated inference with the `Cifar10-R18` model. Each inference happens in a new session. We plot the traces obtained with the same profiler instrumentation. Figure 3b shows the behavior of an NVidia P100 GPU. For 33 inferences on the same model and data, we observe 6 different paths resulting in 4 EQCs. Observe that up to three

---

[3]This approach is also called "auto-tuning" in the literature, e. g., [14; 31].

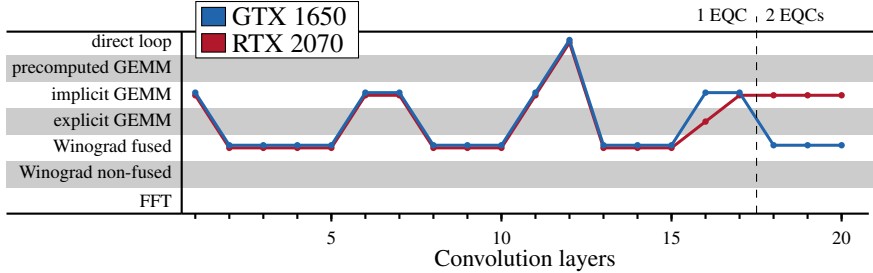

(a) Different GPUs choose different algorithms in certain layers. All GEMM variants produce identical outputs.

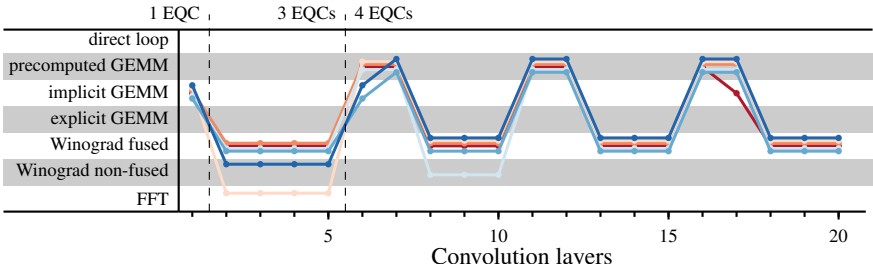

(b) The same GPU chooses different algorithms depending on variance in the microbenchmarks.

Figure 3: Choice of convolution algorithm per layer. Function calls are grouped by approach.

different approaches are used to compute the same convolution (layers 2–5). The leftmost dashed line provides evidence that the intermediate results differ at that layer and propagate further. Another separation happens at layer 6, whereas the choice of algorithm at layer 17 does not fork out another EQC. We confirm from all other traces that this pattern is typical for indeterministic deviations between GPUs.

The host system of the GPU can also influence the race and thus the number of unique algorithm sequences. The red trace in Figure 3a shows one of two unique algorithm sequences for an RTX 2070, which both lead to one deterministic output. For 33 inferences on the same model and data, we deterministically observe two EQCs, one for each GPU. We find that the same GPU can produce different algorithm sequences, resulting in a different number of EQCs on other host systems.

### 3.3 Model-specific influences

The architecture, parameters, and input dimensions of an ML model can influence the number and nature of deviations by amplifying or suppressing influences.

**Number of multiplications/parameters** Numerical deviations are caused by arithmetic operations, therefore more operations mean potentially more deviations. Both larger inputs and a higher number of parameters increase the number of computations during inference. Moreover, the problem dimensions impact the performance of convolution algorithms and affect the algorithm choice.

As deviations arise from rounding errors and different aggregation orders, we hypothesize that more multiplications per convolution layer lead to more deviations. We investigate this with an ablation study for the parameters of standard 2-dimensional convolution: kernel size $k$, number of filters $f$, and input dimensions $i \times j \times c$. Our experiment covers several orders of magnitude in the number of multiplications, ranging from $10^3$ to $10^8$, which is more than the number of multiplications in our Cifar10-R18 model. For each order of magnitude, we fix $k$, $f$ and $c$, and adjust $i$ and $j$ to reach the desired number of multiplications while aiming for square inputs, i.e., $i \approx j$. Figure 4 shows $c = 3$, but other values give the same results. Weights are drawn from a uniform distribution over $[0, 1)$.[4]

Our remaining precision metric (cf. Section 2) measures the magnitude of the deviations. Figure 4a shows that the remaining precision decreases with the number of multiplications and does not vary

---

[4]This does not affect results, as discussed in the "Parameter and input distribution" paragraph of Section 3.5.

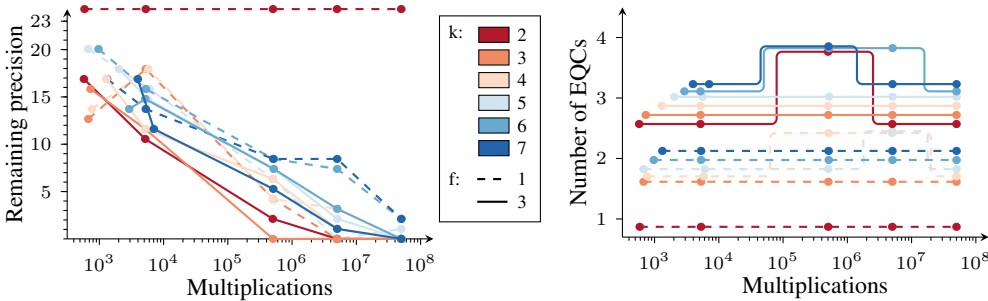

(a) Remaining precision over # of multiplications.

(b) Number of EQCs for the same platforms.

Figure 4: Effect of different convolution dimensions (mapped to the number of multiplications).

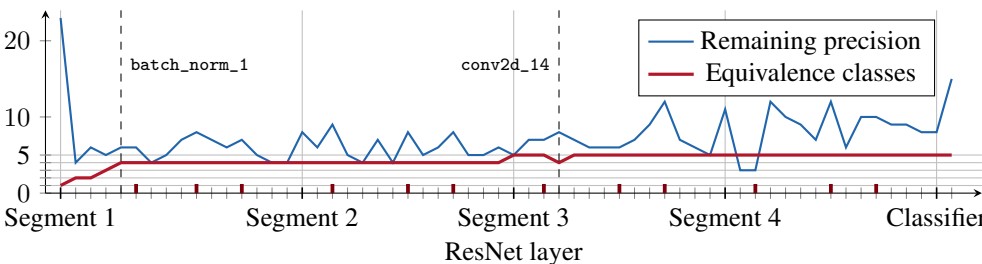

Figure 5: Causes and effects by layer: number of EQCs and remaining precision for `Cifar10-R18`.

significantly between the platforms (excluding the smallest case $k = 2, f = 1$, where no deviations emerge). This is plausible, as approximately Gaussian deviations lead to a lower minimum remaining precision if more independent realizations are aggregated. The remaining precision drops to $0$, which means that at least one deviation is at least as large as factor 2.

Figure 4b shows the number of equivalence classes as a function of the number of multiplications. Observe that the relation is not monotonic. Both $f = 1$ and $f = 3$ produce more EQCs for $10^6$ to $10^7$ multiplications than for $10^8$. The number of filters seems to have a larger influence than the filter size. Note that the maximum number of EQCs in this experiment is four, whereas we observe up to five EQCs for the same set of platforms in our main experiment with complete models. We interpret this as evidence that deviations propagate between layers and may produce forks into different EQCs in later layers.

**Architecture and layer types**    The model architecture defines the operations and hence determines if a cause of deviation is present or not. In particular, convolutional and other parallel data processing layers tend to introduce deviations. Layers that aggregate results and reduce information, such as pooling layers, may reduce or eliminate deviations from preceding layers. This extends to activation functions. Continuous functions that preserve information (e. g., sigmoid and softmax) can maintain or potentially amplify deviations. Functions that reduce information, like the rectified linear unit (ReLU), can have a diminishing effect on deviations. This means that a sufficiently large model can introduce deviations in earlier layers, and remove them in pooling and activation layers. Skip layers can preserve deviations by bypassing information reduction.

To investigate the influence of different layer types in a full inference computation, we instrument a model to output the intermediate results of all layers in addition to the final class label. We verify that this does not alter the EQCs. Figure 5 plots the number of EQCs as well as the remaining precision over the outputs of the 60 layers of `Cifar10-R18`. Note that activations count as distinct layers, marked with a thick dark-red tick, and not every convolution is immediately followed by an activation. The annotated segments on the x-axis refer to the ResNet convention of bundling convolutions of the same size into segments [15].

The remaining precision changes after every layer. This can be explained by the fact that the very first convolution following the input layer already produces deviations that result in multiple EQCs.

As long as more than one EQC exists (i. e., for all remaining layers), the remaining precision indicates how far these EQCs fall apart in the most extreme case. In terms of magnitude, the remaining precision never reaches zero. This corroborates the results of our ablation study (cf. Figure 4a) as the maximum number of multiplications per layer is in the order of $10^6$. Although the number of total multiplications increases monotonically, the ReLU activations increase the remaining precision, visible as peaks of the remaining precision. One instance of aggregation is very pronounced at the rightmost layer, where the output is projected to a small label space.

Turning to the number of EQCs, new ones only emerge as outputs of convolutional and batch normalization layers. The number of EQCs tends to increase over the execution of the model. However, we also observe a reduction in the number of EQCs, which occurs once in a convolutional layer (`conv2d_14`) in the third segment of the model. We did not expect to sees this, in particular given that this layer does not perform any quantization or aggregation. This shows that past deviations may be cancelled out. Another unexpected finding is the emergence of the fourth EQC, which occurs in a batch normalization layer (`batch_norm_1`), and deviations arise in layers other than convolution.

## 3.4 Influence of floating point precision

Figure 6 shows how casting a trained model to a different floating-point precision affects deviations. The figure shows similarities between the EQCs as dendrograms, with remaining precision used as distance metric. Branch lengths within the dendrograms are proportional to the remaining precision between the EQCs, and connections further to the right indicate higher remaining precision. Branches extend past the maximum remaining precision of the format to indicate platforms that produce identical outputs and thus are in the same EQC.

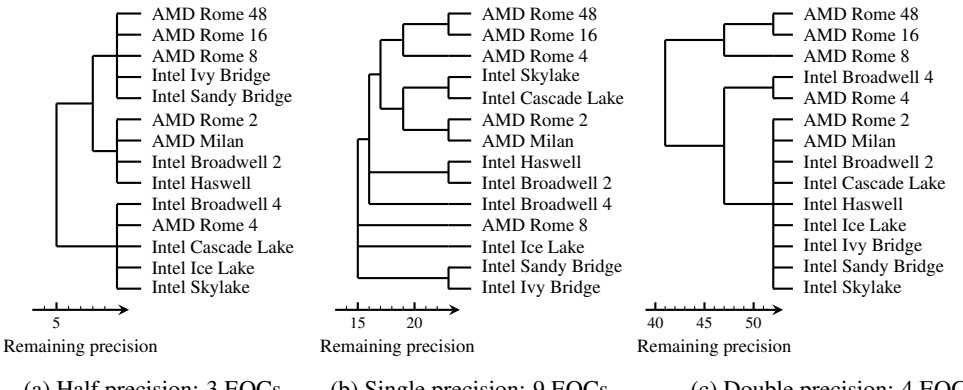

(a) Half precision: 3 EQCs.  (b) Single precision: 9 EQCs.  (c) Double precision: 4 EQCs.

Figure 6: Influence of casting the `Cifar10-R18` model to different floating-point precisions. Both half (float16) and double precision (float64) floating-point numbers generate fewer deviations than single precision. The same pattern holds for all models, cf. Section E in the supplementary material.

Rounding to IEEE-754 16-bit half-precision (`float16`) suppresses deviations and reduces the number of EQCs. However, which EQCs fall together seems rather chaotic in our experiments. For example, AMD Rome processors fall into all three EQCs, depending on their core count. Reducing precision thus mitigates deviations, but in potentially unpredictable ways. Alternatively, the model can be converted to fixed point integer arithmetic, which reportedly eliminates all deviations [34].

Interestingly, increasing precision also reduces the number of EQCs. While we observe a total of four different EQCs when inferring with our model in 64-bit double precision, this is still fewer than for 32-bit single precision. As we simply cast the model to the higher precision without modifying the weights, the lower part of the 64-bit mantissas are zero. This provides more room for shifting and prevents swamping to some extent. Platforms from different EQCs for 32 bit match for 64 bit according to their core count, indicating that deviations due to task parallelism persist. We conjecture that zeroing the last $n$ mantissa bits has the same effect, reducing the performance cost of the mitigation measure at the cost of reduced fidelity.

### 3.5 Potential influences not observed in experiments

Our research identified a number of potential influences that did not surface in isolated experiments.

**Compute graph optimization**   Recent approaches go beyond optimized algorithms and instead transform the operations and their ordering (the compute graph) [23]. For example, the Accelerated Linear Algebra (XLA) project is a JIT compiler for ML operations [39]. The resulting compute graph is specific to the hardware [34]. It determines if and how deviations caused in the hardware come to effect. XLA is not yet used by default [39]. As we did not activate it, we can rule out special compute-graph optimizations as cause for deviations.

**Device placement**   Most ML frameworks allow running operations on both CPU and GPU. In theory this lets them choose the target device at runtime. Under certain conditions, the framework may decide to execute operations on the CPU even though a GPU is available [37]. Since CPU and GPU implementations very likely differ in algorithm, floating-point precision, and aggregation order, the device placement can cause deviations. We log the device placement in our experiments and verify that EQCs do not change with our instrumentation.

**Race conditions**   Task-parallel execution can lead to different aggregation orders depending on the implementation. If all intermediate results are stored in an array and then summed up (e.g., with sum–reduce), the execution time of individual tasks will not change the aggregation order. However, if the results are summed up immediately, with some sort of locking mechanism, the execution time of individual tasks determines the aggregation order. In our experiments, all model outputs on CPUs and GPUs are consistent and deterministic when controlling for the influences we discovered, ruling out race conditions as causes of deviations.

**Parameter and input distributions**   The distributions of model parameters and input values can both affect the shape and magnitude of deviations. To measure their influence, we first extract the `conv2d_11` layer from our `Cifar10-R18` model and capture its intermediate input during a forward pass. Then, we perform inference on twelve combinations of input and parameter distributions on all dual-core GCP instances. We use the original weights, weights drawn from the Glorot uniform distribution [12], and weighs drawn from a Gaussian with $\mu = -0.019, \sigma = 0.226$, fitted to the trained weights. For the inputs we use the captured values, two random permutations of these values (preserving the marginal distribution), and random inputs drawn uniformly from $[0, 1)$. The EQCs for all cases are identical. This suggest that the emergence of EQCs is independent of the parameter and input distribution. Incidentally, the microbenchmarks for algorithm choice on GPUs make a similar assumption. Their inputs are filled with random data.

## 4   Discussion

**Impact**   The deviations we observe might impair ML reproducibility and security. Concerning reproducibility, as different platforms can produce different outputs from the exact same model and input data, it is unlikely that high-precision numerical results can be reproduced exactly. Assertions in automated software tests are prone to fail on specific hardware. Such limitations to reproducibility are not always visible in headline indicators because classifier performance is typically measured on the level of labels. The deviations we study rarely cause label flips for natural inputs. However, we have demonstrated in [35] that label flips can be provoked by searching for synthetic inputs that map to the space between decision boundaries of different platforms. Moreover, Casacuberta et al. [8] point out that numerical deviations can undermine security guarantees of ML implementations with differential privacy. Differences between symbolic math and actual floating-point computations at limited precision have been used to fool ML verifiers [42; 17]. Varying algorithm choices break these assumptions even further, as they fundamentally change the type and order of arithmetic operations. These examples highlight the security impact of our observation. Our study of causes breaks ground for a principled approach to assess and mitigate these risks.

Federated learning distributes ML training tasks over many machines under the assumption of compatible gradients. The systematic deviations observed here call for robustness checks when combining gradients from heterogeneous hardware [7; 25]. The fact that deviations leave traces of the executing hardware enables forensicability and attribution of ML outputs [34]. This could be used,

e. g., in the combat against harmful generated content; but can also be misused, e. g., for unauthorized surveillance. This adds a new aspect to the debate on societal and ethical aspects of ML.

**Mitigation strategies**    As shown in Figure 6, quantization can suppress deviations and reduce the number of EQCs. However, these measures do not mitigate deviations due to algorithm choice, and thus are not applicable to GPUs. While researchers can instruct the ML framework to use deterministic implementations of operators,[5] the fact that different GPUs support different algorithms can lead to deviations even when such options are set. Similarly, TensorCores [28] *may* be used if available, which can also lead to deviations. While they can be disabled, their use cannot be enforced. Reaching full reproducibility requires giving researchers access to low-level functionalities and providing them with fine-grained control over optimizations. Another approach is to sidestep platform-specific optimizations at runtime by transforming the model into a flat compute graph. TFLite, for example, does this by default; yet for the purpose of a smaller memory footprint rather than consistency.

While all these strategies are heuristic, developers of future ML frameworks and accelerator libraries should consider supporting an option for fully deterministic and consistent computation. This comes almost unavoidably at the cost of lower performance. Cryptographic libraries, for example, went a similar path when constant-time options were added in response to the discovery of side-channel attacks [20]. In both domains, successful mitigation depends on exact knowledge of the entire stack down to the hardware.

**Limitations**    While our work covers a lot of ground concerning numerical deviations, there are still some areas for further exploration. Our experiments have intentionally kept some factors constant: the versions of TensorFlow (2.5.0), all dependent libraries and drivers, the compiler version and options used to build TensorFlow, as well as the concurrent load on the system besides our experiments. Additional layer types and other frameworks can also be subject to follow-up work.

## 5    Conclusion

Performing inference on the same trained model and input data is not always consistent across platforms, and sometimes not even deterministic on the same platform. This paper explores the causes for the numerical deviations, in particular in convolution operations. They include the choice of the specific convolution algorithm, the floating-point precision, and the order of aggregation. The order of aggregation is defined by SIMD capabilities and the number of cores of the executing CPU. As the number of CPU cores grows, task parallelism can supersede the deviations caused by SIMD capabilities. GPUs use microbenchmarks to select the fastet supported convolution algorithm at runtime. We find that this can produce seemingly indeterministic behavior when different algorithms are chosen in different sessions. We validate our findings on 75 platforms, including CPUs and GPUs, hosted locally and on two large commercial cloud platforms. Our measurement infrastructure, which is made available on GitHub,[6] can facilitate future comparative studies. Our analysis offers a number of mitigation strategies, but certain deviations appear unavoidable at present. These findings imply that authors interested in improving reproducibility should meticulously document specifics of the computation hardware, in addition to the amount of compute [27], and provide high-precision intermediate results as reference points for replication attempts.

## Acknowledgments and Disclosure of Funding

We thank Matthias Kramm for insightful comments on earlier versions of this paper. We also thank Michael Fröwis, Jakob Hollenstein, Martin Beneš, and Patrik Keller for fruitful discussions during this research. Parts of this work were funded by the European Union's Horizon 2020 research and innovation programme under grant agreement No. 101021687, and a PhD Fellowship from NEC Laboratories Europe (2021–2022).

---

[5]For example, by setting the `enable_op_determinism` flag, TensorFlow chooses the first available convolution algorithm rather than the fastest one.

[6]`https://github.com/uibk-iNNference/`

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
