# Supplementary Material
## Causes and Effects of Unanticipated Numerical Deviations in Neural Network Inference Frameworks

**Alexander Schlögl**  **Nora Hofer**  **Rainer Böhme**

Department of Computer Science
Universität Innsbruck, 6020 Innsbruck, Austria

{`alexander.schloegl` | `nora.hofer` | `rainer.boehme`} @ `uibk.ac.at`

## A  Detailed results

Table SUP-1 shows the CPU EQCs in full detail, including information on hardware, model, and input. Table SUP-2 shows the same information for GPUs. This data is the source for Figure 1 in the main paper.

Table SUP-3 lists all observed CPU flags and their corresponding cluster index. Some flags were present on all machines, and were thus filtered from Table 1 in the main paper. These are marked with *C*, for common.

## B  Methodology

Our experiments require instrumentation at various levels of the ML software stack, shown in Figure SUP-1. The interfaces to the ML framework use different programming languages. Parts of the stack are not accessible for analysis (e. g., microcode on CPUs, vendor libraries for GPUs).

**Information extraction**   We captured as much information about the entire inference pipeline as possible, using TensorFlow's own profiler. It includes underlying tools like NVidia's `nvprof` profiler, and allows us to investigate function calls in the accelerator libraries. Information about the computing devices is taken from `/proc/cpuinfo` and the TensorFlow device information, respectively. For CPUs we fill the microarchitecture field by cross-referencing the `family` and `model` fields of the CPUID. GPUs are uniquely identifiable by their names, including the microarchitecture. Device names are given with as much detail as provided by the machine; due to shared tenancy, device information for cloud CPUs may be reported with less detail. Memory sizes are taken from the `psutil` Python module for CPUs, and from the TensorFlow reported `memory_limit` for GPUs.

**Containerization**   To ensure that the same experiments are run on a large number of cloud instances, we use Docker to fix the software environment and package versions. The Docker image is built locally and uploaded to image registries of both cloud providers used. From there, the image is pulled to the respective target machines and used to run the experiments.

Our experiments run in a Docker container based on the `tensorflow:2.5.0-gpu` image. This image runs Python version 3.6.9, and we additionally install `clang` version 6.0.0. The TensorFlow version is 2.5.0. In this version, XLA is not enabled by default, and was not explicitly activated. The container already handles the GPU setup, and no additional steps are necessary. For the cloud instances with GPUs, we forward them to the container using Docker's `--gpus` flag.

Table SUP-1: Full results for CPU instances. EQCs are assigned increasing integers from top to bottom. Table cells identical with their left neighbor are slightly faded. CPUs are separated into CPU classes (CCs) based on available x86 extensions and clustered core count. The first occurrence of an EQC per column is marked in bold.

| | | | | | | | | Dataset | CIFAR-10 | | | | | DeepWeeds | | |
| | | | | | | | | Model size | Small | | | Medium | | | Large | | |
| | | | | | | | | Sample index | 0 | 1 | 6 | 0 | 1 | 6 | 0 | 1 | 6 |
| | Vendor | Cores | Generation | Device name | Mem. | Cloud | CC | | | | | | | | | | |
|---|---|---|---|---|---|---|---|---|---|---|---|---|---|---|---|---|---|
| (1) | AMD | 2 | Milan | 7B13 | 7.8 | GCP | 0 | | **1** | 1 | 1 | 1 | 1 | 1 | 1 | 1 | 1 |
| | AMD | 2 | Rome | 7B12 | 7.8 | GCP | 0 | | 1 | 1 | 1 | 1 | 1 | 1 | 1 | 1 | 1 |
| | AMD | 4 | Milan | 7B13 | 15.6 | GCP | 1 | | 1 | 1 | 1 | **2** | **2** | **2** | **2** | **2** | **2** |
| | AMD | 4 | Rome | 7B12 | 15.6 | GCP | 1 | | 1 | 1 | 1 | 2 | 2 | 2 | 2 | 2 | 2 |
| | AMD | 8 | Milan | 7B13 | 31.4 | GCP | 2 | | 1 | 1 | 1 | **3** | **3** | **3** | **3** | **3** | **3** |
| (6) | AMD | 8 | Rome | 7B12 | 31.4 | GCP | 2 | | 1 | 1 | 1 | 3 | 3 | 3 | 3 | 3 | 3 |
| | AMD | 16 | Milan | 7B13 | 62.8 | GCP | 3 | | 1 | 1 | 1 | **4** | **4** | **4** | **4** | **4** | **4** |
| | AMD | 16 | Rome | 7B12 | 62.8 | GCP | 3 | | 1 | 1 | 1 | 4 | 4 | 4 | 4 | 4 | 4 |
| | AMD | 32 | Milan | 7B13 | 125.9 | GCP | 4 | | 1 | 1 | 1 | 4 | 4 | 4 | **5** | **5** | **5** |
| | AMD | 32 | Rome | 7B12 | 125.9 | GCP | 4 | | 1 | 1 | 1 | 4 | 4 | 4 | 5 | 5 | 5 |
| (11) | Intel | 2 | Sandy Br. | Xeon | 7.3 | GCP | 5 | | **2** | **2** | **2** | **5** | **5** | **5** | **6** | **6** | **6** |
| | Intel | 2 | Ivy Br. | Xeon | 7.3 | GCP | 5 | | 2 | 2 | 2 | 5 | 5 | 5 | 6 | 6 | 6 |
| | Intel | 2 | Haswell | Xeon | 7.3 | GCP | 6 | | 1 | 1 | 1 | **6** | **6** | **6** | **7** | **7** | **7** |
| | Intel | 2 | Haswell | E5-2676 | 3.8 | AWS | 6 | | 1 | 1 | 1 | 6 | 6 | 6 | 7 | 7 | 7 |
| | Intel | 2 | Broadwell | Xeon | 7.3 | GCP | 6 | | 1 | 1 | 1 | 6 | 6 | 6 | 7 | 7 | 7 |
| (16) | Intel | 2 | Broadwell | E5-2686 | 7.8 | AWS | 6 | | 1 | 1 | 1 | 6 | 6 | 6 | 7 | 7 | 7 |
| | Intel | 2 | Skylake | 8175M | 15.3 | AWS | 7 | | **3** | **3** | **3** | **7** | **7** | **7** | **8** | **8** | **8** |
| | Intel | 2 | Skylake | Xeon | 7.8 | GCP | 7 | | 3 | 3 | 3 | 7 | 7 | 7 | 8 | 8 | 8 |
| | Intel | 2 | Skylake | Xeon | 7.3 | GCP | 7 | | 3 | 3 | 3 | 7 | 7 | 7 | 8 | 8 | 8 |
| | Intel | 2 | Skylake | 8175M | 7.5 | AWS | 7 | | 3 | 3 | 3 | 7 | 7 | 7 | 8 | 8 | 8 |
| (21) | Intel | 2 | Skylake | 8259CL | 7.6 | AWS | 7 | | 3 | 3 | 3 | 7 | 7 | 7 | 8 | 8 | 8 |
| | Intel | 2 | Skylake | 8259CL | 15.3 | AWS | 7 | | 3 | 3 | 3 | 7 | 7 | 7 | 8 | 8 | 8 |
| | Intel | 2 | Skylake | 8259CL | 7.7 | AWS | 7 | | 3 | 3 | 3 | 7 | 7 | 7 | 8 | 8 | 8 |
| | Intel | 2 | Skylake | 8151 | 15.3 | AWS | 7 | | 3 | 3 | 3 | 7 | 7 | 7 | 8 | 8 | 8 |
| | Intel | 2 | Ice Lake | Xeon | 7.8 | GCP | 8 | | 3 | 3 | 3 | **8** | **8** | **8** | 8 | 8 | 8 |
| (26) | Intel | 4 | Sandy Br. | Xeon | 14.7 | GCP | 9 | | 2 | 2 | 2 | **9** | **9** | **9** | 6 | 6 | 6 |
| | Intel | 4 | Ivy Br. | Xeon | 14.7 | GCP | 9 | | 2 | 2 | 2 | 9 | 9 | 9 | 6 | 6 | 6 |
| | Intel | 4 | Haswell | E5-2666 | 7.3 | AWS | 10 | | 1 | 1 | 1 | **10** | **10** | **10** | 7 | 7 | 7 |
| | Intel | 4 | Haswell | Xeon | 14.7 | GCP | 10 | | 1 | 1 | 1 | 10 | 10 | 10 | 7 | 7 | 7 |
| | Intel | 4 | Haswell | E5-2676 | 15.6 | AWS | 10 | | 1 | 1 | 1 | 10 | 10 | 10 | 7 | 7 | 7 |
| (31) | Intel | 4 | Haswell | E7-8880 | 119.9 | AWS | 10 | | 1 | 1 | 1 | 10 | 10 | 10 | 7 | 7 | 7 |
| | Intel | 4 | Broadwell | Xeon | 14.7 | GCP | 10 | | 1 | 1 | 1 | 10 | 10 | 10 | 7 | 7 | 7 |
| | Intel | 4 | Skylake | 8124M | 7.4 | AWS | 11 | | 3 | 3 | 3 | **11** | **11** | **11** | 8 | 8 | 8 |
| | Intel | 4 | Skylake | 8275CL | 7.5 | AWS | 11 | | 3 | 3 | 3 | 11 | 11 | 11 | 8 | 8 | 8 |
| | Intel | 4 | Skylake | 8124M | 9.9 | AWS | 11 | | 3 | 3 | 3 | 11 | 11 | 11 | 8 | 8 | 8 |
| (36) | Intel | 4 | Skylake | Xeon | 15.6 | GCP | 11 | | 3 | 3 | 3 | 11 | 11 | 11 | 8 | 8 | 8 |
| | Intel | 4 | Skylake | Xeon | 14.7 | GCP | 11 | | 3 | 3 | 3 | 11 | 11 | 11 | 8 | 8 | 8 |
| | Intel | 4 | Ice Lake | Xeon | 15.6 | GCP | 12 | | 3 | 3 | 3 | 11 | 11 | 11 | 8 | 8 | 8 |
| | Intel | 8 | Sandy Br. | Xeon | 29.4 | GCP | 13 | | 2 | 2 | 2 | **12** | **12** | **12** | 6 | 6 | 6 |
| | Intel | 8 | Ivy Br. | Xeon | 29.4 | GCP | 13 | | 2 | 2 | 2 | 12 | 12 | 12 | 6 | 6 | 6 |
| (41) | Intel | 8 | Haswell | Xeon | 29.4 | GCP | 14 | | 1 | 1 | 1 | **13** | **13** | **13** | 7 | 7 | 7 |
| | Intel | 8 | Broadwell | Xeon | 29.4 | GCP | 14 | | 1 | 1 | 1 | 13 | 13 | 13 | 7 | 7 | 7 |
| | Intel | 8 | Skylake | Xeon | 31.4 | GCP | 15 | | 3 | 3 | 3 | **14** | **14** | **14** | 8 | 8 | 8 |
| | Intel | 8 | Skylake | Xeon | 29.4 | GCP | 15 | | 3 | 3 | 3 | 14 | 14 | 14 | 8 | 8 | 8 |
| | Intel | 8 | Coffee Lake | i7-9700 | 31.2 | local | 16 | | 1 | 1 | 1 | 13 | 13 | 13 | **9** | **9** | **9** |
| (46) | Intel | 8 | Coffee Lake | E3-1270 | 31.3 | local | 16 | | 1 | 1 | 1 | 13 | 13 | 13 | 9 | 9 | 9 |
| | Intel | 8 | Ice Lake | Xeon | 31.4 | GCP | 17 | | 3 | 3 | 3 | 14 | 14 | 14 | 8 | 8 | 8 |
| | Intel | 12 | Ivy Br. | i7-4930K | 62.8 | local | 18 | | 2 | 2 | 2 | **15** | **15** | **15** | **10** | **10** | **10** |
| | Intel | 12 | Ivy Br. | i7-4930K | 3.8 | local | 18 | | 2 | 2 | 2 | 15 | 15 | 15 | 10 | 10 | 10 |
| | Intel | 16 | Sandy Br. | Xeon | 58.9 | GCP | 19 | | 2 | 2 | 2 | 15 | 15 | 15 | **11** | **11** | **11** |
| (51) | Intel | 16 | Ivy Br. | Xeon | 58.9 | GCP | 19 | | 2 | 2 | 2 | 15 | 15 | 15 | 11 | 11 | 11 |
| | Intel | 16 | Haswell | Xeon | 58.9 | GCP | 20 | | 1 | 1 | 1 | **16** | **16** | **16** | **12** | **12** | **12** |
| | Intel | 16 | Broadwell | Xeon | 58.9 | GCP | 20 | | 1 | 1 | 1 | 16 | 16 | 16 | 12 | 12 | 12 |
| | Intel | 16 | Skylake | Xeon | 62.8 | GCP | 21 | | 3 | 3 | 3 | 16 | 16 | 16 | **13** | 12 | **13** |
| | Intel | 16 | Skylake | Xeon | 58.9 | GCP | 21 | | 3 | 3 | 3 | 16 | 16 | 16 | 13 | 13 | 13 |
| (56) | Intel | 16 | Ice Lake | Xeon | 62.8 | GCP | 22 | | 3 | 3 | 3 | 16 | 16 | 16 | 13 | 13 | 13 |
| | Intel | 32 | Sandy Br. | Xeon | 117.9 | GCP | 23 | | 2 | 2 | 2 | 15 | 15 | 15 | **14** | **14** | **14** |
| | Intel | 32 | Ivy Br. | Xeon | 117.9 | GCP | 23 | | 2 | 2 | 2 | 15 | 15 | 15 | 14 | 14 | 14 |
| | Intel | 32 | Haswell | Xeon | 117.9 | GCP | 24 | | 1 | 1 | 1 | 16 | 16 | 16 | **15** | **15** | **15** |
| | Intel | 32 | Broadwell | Xeon | 117.9 | GCP | 24 | | 1 | 1 | 1 | 16 | 16 | 16 | 15 | 15 | 15 |
| (61) | Intel | 32 | Skylake | Xeon | 125.9 | GCP | 25 | | 3 | 3 | 3 | 16 | 16 | 16 | **16** | **16** | **16** |
| | Intel | 32 | Skylake | Xeon | 117.9 | GCP | 25 | | 3 | 3 | 3 | 16 | 16 | 16 | 16 | 16 | 16 |
| | Intel | 32 | Ice Lake | Xeon | 125.8 | GCP | 26 | | 3 | 3 | 3 | 16 | 16 | 16 | 16 | 16 | 16 |
| | Intel | 48 | Skylake | 8275CL | 92.2 | AWS | 25 | | 3 | 3 | 3 | 16 | 16 | 16 | 16 | 16 | 16 |

Table SUP-2: Full results for GPU instances. EQCs are assigned increasing integers from top to bottom. Table cells identical with their left neighbor are slightly faded. The first occurrence of an EQC per column is marked in bold. Cells marked with an asterisk (*) indicate that indeterminsm was observed. The equivalence class is based on the most frequently observed output.

| | | Dataset | | CIFAR-10 | | | | | | DeepWeeds | | |
| | | Model size | | Small | | | Medium | | | Large | | |
| | | Sample index | | 0 | 1 | 6 | 0 | 1 | 6 | 0 | 1 | 6 |
| Vendor | Generation | Device name | Cloud | | | | | | | | | |
| NVidia | Kepler | K80 | GCP | 1 | 1 | 1 | 1* | 1* | 1* | 1 | 1 | 1 |
| NVidia | Maxwell | GTX 970 | local | 2* | 2 | 2 | 2 | 2 | 2* | 2 | 2 | 2 |
| NVidia | Maxwell | GTX 980 | local | 3* | 3 | 3 | 3 | 3 | 3 | 3 | 3 | 3 |
| NVidia | Maxwell | M60 | AWS | 3 | 3 | 3 | 3* | 4 | 4* | 4 | 4 | 4 |
| NVidia | Pascal | P100 | GCP | 4* | 4* | 4* | 4* | 5* | 5* | 5 | 5 | 5 |
| NVidia | Volta | V100 | GCP | 5* | 5* | 5* | 5* | 6* | 6* | 6* | 6* | 6* |
| NVidia | Turing | GTX 1650 | local | 6 | 6 | 6 | 6 | 7 | 7 | 7 | 7 | 7 |
| NVidia | Turing | RTX 2070 | local | 7 | 7 | 7 | 7 | 8 | 8* | 8 | 8 | 8 |
| NVidia | Turing | T4 | AWS | 7* | 7* | 7* | 8 | 9 | 9 | 9* | 9* | 9 |
| NVidia | Turing | T4 | GCP | 7 | 7 | 7 | 8* | 9* | 9 | 9* | 9* | 9* |
| NVidia | Ampere | A100 | GCP | 8 | 8 | 8 | 9 | 10* | 10* | 10* | 10* | 10* |

To ensure a clean tensor graph for each experiment, we create a new Python session for each inference. To this end we create a small CLI application that takes the model name and input path as arguments and computes the inference inside the container.

**Dead ends** In additional but eventually unsuccessful steps, we aimed to fully understand the paths taken during execution by instrumenting TensorFlow with the `gdb` debugger, as well as the `perf` and `valgrind` profiling tools. We specifically hoped that `valgrind`'s [7] cachegrind tool would provide insight into the actual code executed on CPU, but the tremendous amount of inlining in TensorFlow's codebase yielded no usable results. Outputs from `perf` were too noisy to be useful. TensorFlow's codebase was too large for `gdb` [12] analysis to be useful, both in interactive and automated scenarios. The record-and-replay debugger `rr` [9] could not deal with the complexities of TensorFlow's codebase and could not successfully record a single inference.

## C   Convolution algorithms and functions

We summarize the main approaches for computing convolutions.

**General matrix multiplication (GEMM)** Convolution can be calculated via matrix multiplication by extracting the relevant parts of the image into a Toeplitz matrix, replicating the filter as needed, and multiplying the two matrices. This is equivalent to an unrolled loop variant of the naive implementation with data replication for better access [1]. Even with data structures optimized for sparse matrices, the Toeplitz matrix has many duplicate entries, producing significant memory overhead. A divide-and-conquer alternative reduces the memory overhead by utilizing the fact that matrices are stored in memory as contiguous 1-dimensional arrays [1]. The full convolution is split into multiple smaller convolutions, and the results are reconstructed afterwards. The divide-and-conquer approach causes a different order of aggregations, which can lead to numerically different results.

**Winograd algorithm** Winograd short convolution [3] is a fast convolution algorithm that transforms both inputs and filters to achieve a lower number of multiplications. Its transformation is inherently lossy and involves multiple rounding steps. The Winograd algorithm exists in fused and non-fused variants. The fused variant performs transformation, point-wise multiplication, and inverse transformation in a single step, reducing the number of memory accesses, whereas the non-fused variant performs all steps separately.

Table SUP-3: Full list of CPU flags in alphabetical order and their corresponding cluster index. Flags with cluster index $C$ (common) are present on all CPUs and are not show in the original table.

| Flag | Cluster | Flag | Cluster | Flag | Cluster |
|---|---|---|---|---|---|
| 3dnowext | 3 | fma | 0 | pse36 | $C$ |
| 3dnowprefetch | 7 | fpu | $C$ | pti | 14 |
| abm | 0 | fsgsbase | 10 | rdpid | 3 |
| adx | 7 | fxsr | $C$ | rdrand | 10 |
| aes | $C$ | fxsr_opt | 3 | rdrnd | 10 |
| apic | $C$ | gfni | 2 | rdseed | 7 |
| arat | $C$ | hle | 11 | rdtscp | $C$ |
| arch_capabilities | 4 | ht | $C$ | rep_good | $C$ |
| avx | $C$ | hypervisor | $C$ | rtm | 11 |
| avx2 | 0 | ibpb | $C$ | sep | $C$ |
| avx512_bitalg | 2 | ibrs | $C$ | sha | 6 |
| avx512_vbmi2 | 2 | ibrs_enhanced | 8 | sha_ni | 6 |
| avx512_vnni | 8 | invpcid | 12 | smap | 7 |
| avx512_vpopcntdq | 2 | invpcid_single | 12 | smep | 10 |
| avx512bitalg | 2 | lahf_lm | $C$ | ss | 4 |
| avx512bw | 1 | lm | $C$ | ssbd | $C$ |
| avx512cd | 1 | mca | $C$ | sse | $C$ |
| avx512dq | 1 | mce | $C$ | sse2 | $C$ |
| avx512f | 1 | md_clear | 4 | sse4_1 | $C$ |
| avx512ifma | 2 | misalignsse | 3 | sse4_2 | $C$ |
| avx512vbmi | 2 | mmx | $C$ | sse4a | 3 |
| avx512vbmi2 | 2 | mmxext | 3 | ssse3 | $C$ |
| avx512vl | 1 | movbe | 0 | stibp | $C$ |
| avx512vnni | 8 | mpx | 13 | syscall | $C$ |
| avx512vpopcntdq | 2 | msr | $C$ | topoext | 3 |
| bmi1 | 0 | mtrr | $C$ | tsc | $C$ |
| bmi2 | 0 | nonstop_tsc | $C$ | tsc_adjust | $C$ |
| clflush | $C$ | nopl | $C$ | tsc_known_freq | $C$ |
| clflushopt | 5 | npt | 3 | umip | 6 |
| clwb | 5 | nrip_save | 3 | vaes | 2 |
| clzero | 3 | nx | $C$ | vme | $C$ |
| cmov | $C$ | osvw | 3 | vmmcall | 3 |
| cmp_legacy | 3 | osxsave | $C$ | vpclmulqdq | 2 |
| constant_tsc | $C$ | pae | $C$ | x2apic | 4 |
| cpuid | $C$ | pat | $C$ | xgetbv1 | 5 |
| cr8_legacy | 3 | pcid | 9 | xsave | $C$ |
| cx16 | $C$ | pclmulqdq | $C$ | xsavec | 5 |
| cx8 | $C$ | pdpe1gb | $C$ | xsaveerptr | 3 |
| de | $C$ | pge | $C$ | xsaveopt | $C$ |
| erms | 9 | pni | $C$ | xsaves | 1 |
| extd_apicid | 3 | popcnt | $C$ | xtopology | 4 |
| f16c | 10 | pse | $C$ | | |

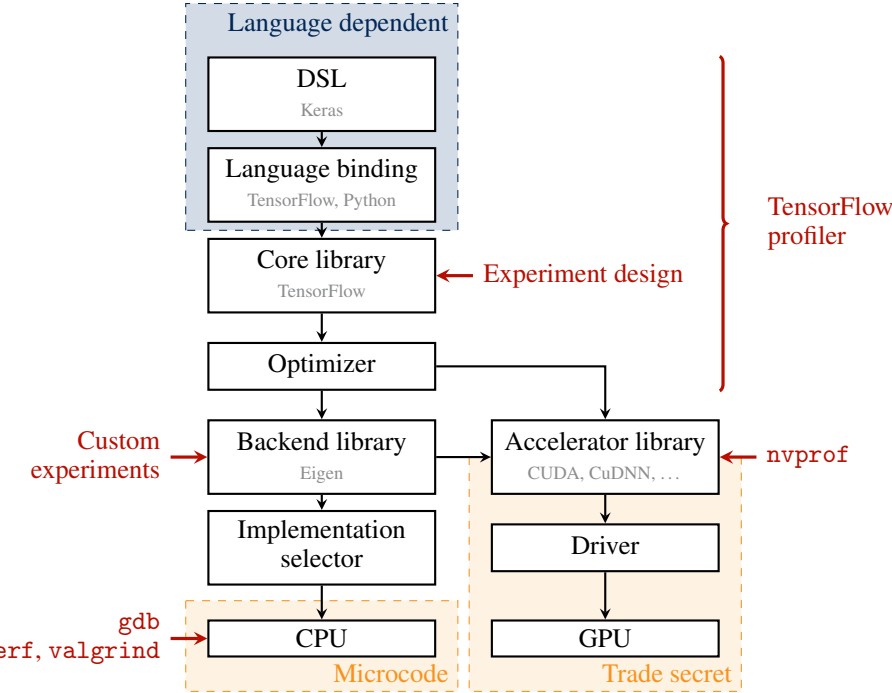

Figure SUP-1: Visualization of the ML software stack. Instrumentation and tools shown in red.

**Fourier transformation approaches**   Approaches using fast Fourier transformation (FFT) exploit the fact that spatial convolution is equivalent to point-wise multiplication in the frequency domain. As FFT is also used in many other contexts, especially in signal processing, optimized implementations are commonly available. The transformation to and from the frequency domain is inherently lossy for finite numerical precision [6].

Most algorithms come in two major variants: an *explicit* variant, where precomputations happen in a separate step, and an *implicit* variant, where precomputations happen right in the algorithm (e. g., input replication for the im2 algorithm). In addition to the above mentioned algorithms, modern GPUs also include special hardware to compute convolution directly [8]. The performance of the above algorithms varies with the use case. Table SUP-4 compares advantages and disadvantages of each approach based on the literature [2].

## D   Datasets and preprocessing

We use the CIFAR-10 [5] and Deep Weeds [10] datasets for our experiments, both obtained from the `tensorflow_datasets` Python package. CIFAR-10 has ten classes and consists of 50 000 training and 10 000 test samples. Deep Weeds has nine classes and consists of 17 509 samples, which we split into training (first 85 %) and test set (final 15 %). We transform all samples from their integer range $[0, 255]$ to floating-point numbers in the range $[0, 1]$ by dividing by 255. All training and test sets are shuffled using the TensorFlow `dataset.shuffle` function with a buffer size of the entire dataset, and random seed 42. We batch the samples with a batch size of 32 for all datasets.

Table SUP-5 provides a summary of all models used in this paper.

For CIFAR-10 we use a small custom CNN with two convolutional layers (similar to the VGG architecture [11]), a ResNet18, and a ResNet50v2 [4] followed by a flatten and dense layer with the required 10 neurons and softmax activation. The small custom CNN is documented in Table SUP-6. For Deep Weeds we use the pre-trained model provided by the authors at `https://github.com/AlexOlsen/DeepWeeds`.

**Training**   The CIFAR-10 models are trained for 30 epochs. The entire training set is used for every epoch. The models were trained on an RTX 3080 GPU.

Table SUP-4: Overview of convolution algorithms. Characteristics based on [2].

| Approach | Time | Memory | Strided | Generation | Name |
|---|---|---|---|---|---|
| direct loop | − | ++ | ++ | Volta | fused conv/ReLU |
|  |  |  |  |  | grouped naive kernel |
| GEMM | + | −− / + | ++ / −− | Ampere | implicit |
|  |  |  |  | Kepler | GEMM |
|  |  |  |  | generic | explicit single precision |
|  |  |  |  | generic | implicit |
|  |  |  |  | generic | precomputed |
| Winograd | ++ | − | − | Ampere | Winograd |
|  |  |  |  | Maxwell | Winograd |
|  |  |  |  | Maxwell | non-fused |
|  |  |  |  | Turing | non-fused |
|  |  |  |  | Volta | compiled |
|  |  |  |  | Volta | non-fused |
| FFT |  | − | + |  | FFT GEMM |

Table SUP-5: Summary of models used.

| Dataset | ResNet[4] | Parameters | | Test set set accuracy |
|---|---|---|---|---|
|  |  | Convolution layers | Total |  |
| CIFAR-10 [5] | 18 | 11,170,816 | 11,191,306 | 60 % |
| DeepWeeds [10] | 50v2 | 23,556,608 | 24,744,457 | 95 % |
| CIFAR-10 | Cifar10-small | Custom (cf. Table SUP-6) | 464 | 59,354 |

Table SUP-6: Summary of model `Cifar10-small`. ReLU activation and max pooling are used except for the experiments in Section E.

| Layer name | Layer type | Output shape | # params |
|---|---|---|---|
| input | InputLayer | $32 \times 32 \times 3$ | 0 |
| conv2d | Conv2D | $32 \times 32 \times 3$ | 84 |
| activation | Activation | $32 \times 32 \times 3$ | 0 |
| pooling2d | Pooling2 | $16 \times 16 \times 3$ | 0 |
| conv2d_1 | Conv2D | $16 \times 16 \times 5$ | 380 |
| activation | Activation | $32 \times 32 \times 3$ | 0 |
| pooling2d_1 | Pooling2 | $8 \times 8 \times 5$ | 0 |
| flatten_1 | Flatten | 320 | 0 |
| dense_1 | Dense | 128 | 41088 |
| dense_2 | Dense | 128 | 16512 |
| dense_3 | Dense | 10 | 1290 |

**Evaluation** The Deep Weeds model reaches 95 % accuracy on the test set, as reported in the original paper [10]. For CIFAR-10, model `Cifar10-small` reaches 53.18 % accuracy, and the `Cifar10-R18` reaches 60.25 % accuracy. These accuracies are not competitive with the state of the art, but sufficiently better than random guessing. We can safely assume that the kernels learn meaningful weights.

**Experiment samples** We process three samples for each of our models to measure the consistency of our results. The first sample is the first test sample (for simplicity); we additionally use a sample from a different class (sample index 1 for CIFAR-10, and index 6 for Deep Weeds), a sample from the same class as the first sample is also used (index 6 for CIFAR-10, and index 1 for Deep Weeds). All sample indexes refer to the unshuffled test set of the respective dataset.

# E   Experiments supporting the rebuttal and author response phase

**Switching precision for all models**

**Request:** "Maybe other Neural Networks could be tested to see if they follow the same pattern for single-point precision."

We repeat the experiments producing Figure 6 in the main paper for both the `Cifar10-small` and `DeepWeeds-R50v2` models. Results for `Cifar10-small` are shown in Figure SUP-2, and results for `DeepWeeds-R50v2` are shown in Figure SUP-3. The figures are structured in the same way as Figure 6 in the main paper.

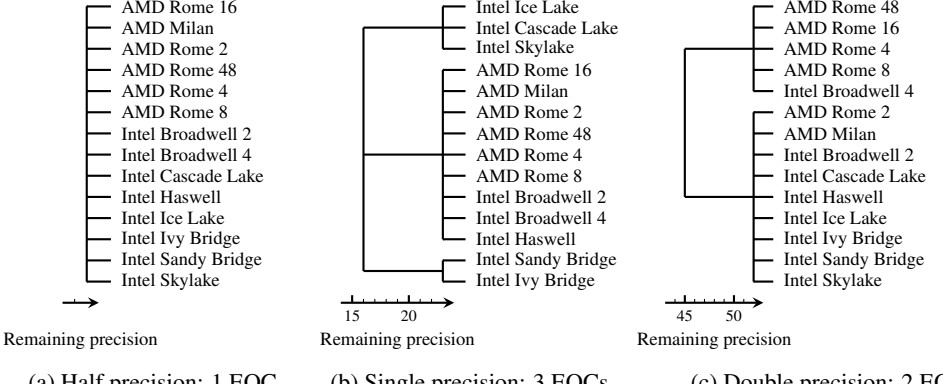

(a) Half precision: 1 EQC.    (b) Single precision: 3 EQCs.    (c) Double precision: 2 EQCs.

Figure SUP-2: Influence of casting the `Cifar10-small` model to different floating-point precisions. Both half and double precision floating-point generate fewer deviations than single precision.

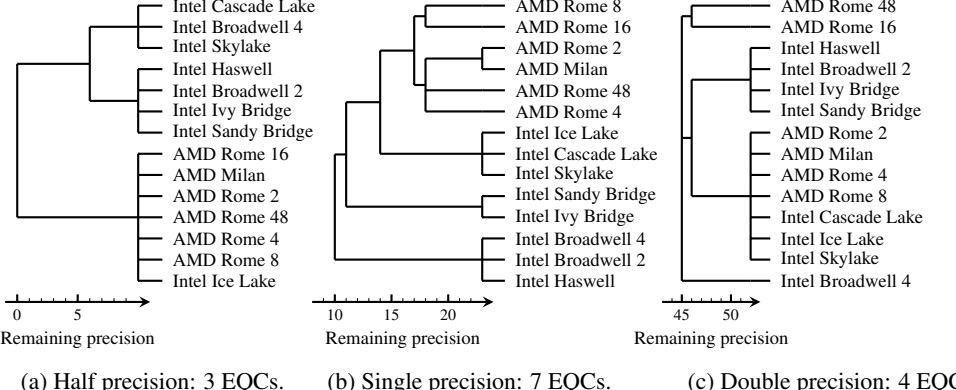

(a) Half precision: 3 EQCs.    (b) Single precision: 7 EQCs.    (c) Double precision: 4 EQCs.

Figure SUP-3: Influence of casting the `DeepWeeds-R50v2` model to different floating-point precisions. Both half and double precision floating-point generate fewer deviations than single precision.

Table SUP-7: Validation accuracy of modified models used for Figures SUP-4 and SUP-5.

| Model name | Activation | Pooling | Validation accuracy |
|---|---|---|---|
| `Cifar10-R18` | Sigmoid | AvgPool | 47.7 % |
| `Cifar10-small` | Sigmoid | AvgPool | 54.6 % |

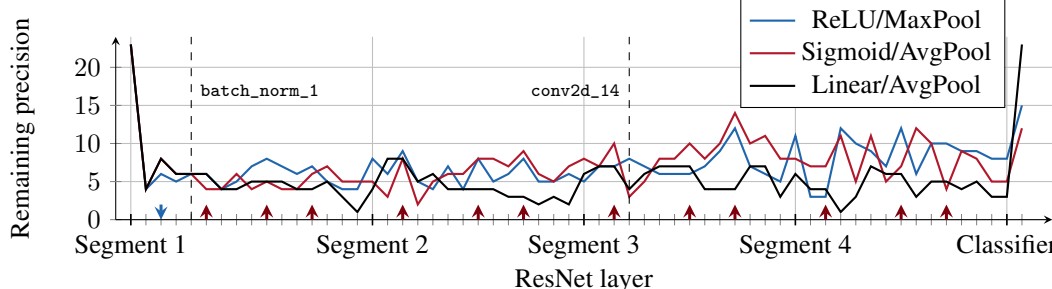

Figure SUP-4: Influence of activation and pooling functions for ResNet-18 architecture. Activation layers are indicated on the x-axis by red upward-facing arrows; pooling layers are indicated by blue downward-facing arrows. Model variants with sigmoid activation use Xavier initialization (`glorot_uniform`).

Again, the number of EQCs is largest for single precision, and decreases for both half and double precision. The distribution of EQCs for single precision is similar to the `Cifar10-R18` model. Changes for double precision are less clear cut, and CPUs with different core counts fall into the same EQC. We conclude that the pattern of EQCs is similar for all models: single precision generates the most EQCs, and half and double precision generate fewer EQCs, but still more than one.

**Weight distribution and stable remaining precision**

**Request:** "can the authors try Xavier initialization + sigmoid activation (instead of He + ReLU in typical Resnet) or replace MaxPool to MeanPool to see if this behavior still holds. This should tell which is the cause of non-diminishing remaining precision."

To answer this question we modify models `Cifar10-small` and `Cifar10-R18` to use sigmoid activation with Xavier initialization for all activation layers. `MaxPool` layers are replaced with `AvgPool`[1] layers. The resulting models are trained for 30 epochs on the CIFAR-10 training set, using the same code as the ReLU models. No tuning of hyperparameters is performed. Table SUP-7 reports the final validation accuracy of the models. As with the ReLU models, this is not competitive with the state of the art, but significantly better than random guessing. Figures SUP-4 and SUP-5 show the results for the modified models. Because the reviewer specifically mentions the initialization we include an untrained version of the model in the results, shown in Figures SUP-6 and SUP-7

We follow the same experimental procedure as in Section 3.3 in the paragraph "Architecture and layer types" to obtain the remaining precision and number of EQCs. Figure SUP-4 show the results for model `Cifar10-R18`. The tick marks indicating the activation layers have been replaced by red upward-facing arrows. Blue downward-facing arrows indicate pooling layers.

The remaining precision for ReLU activation in Figure SUP-4 is the same as in Figure 5 in the main paper, and activation layers either increase the remaining precision or leave it unaffected. In contrast, the first and last sigmoid activation layers decrease the remaining precision. The remaining sigmoid activation layers also either increase the remaining precision or leave it unaffected, same as the ReLU activation layers.

The single pooling layer after the first convolution increases the remaining precision for both `MaxPool` and `AvgPool`. The number of EQCs is the same for both variants of the model, and is not shown to save space.

---

[1]We stay consistent with TensorFlow naming and refer to MeanPool as AvgPool.

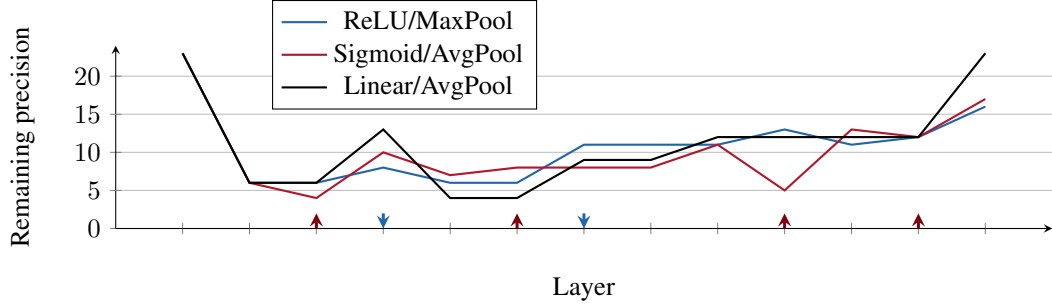

Figure SUP-5: Influence of activation and pooling functions for `Cifar10-small` (cf. Table SUP-6). Activation layers are indicated on the x-axis by red upward-facing arrows; pooling layers are indicated by blue downward-facing arrows. Model variants with sigmoid activation use Xavier initialization (`glorot_uniform`).

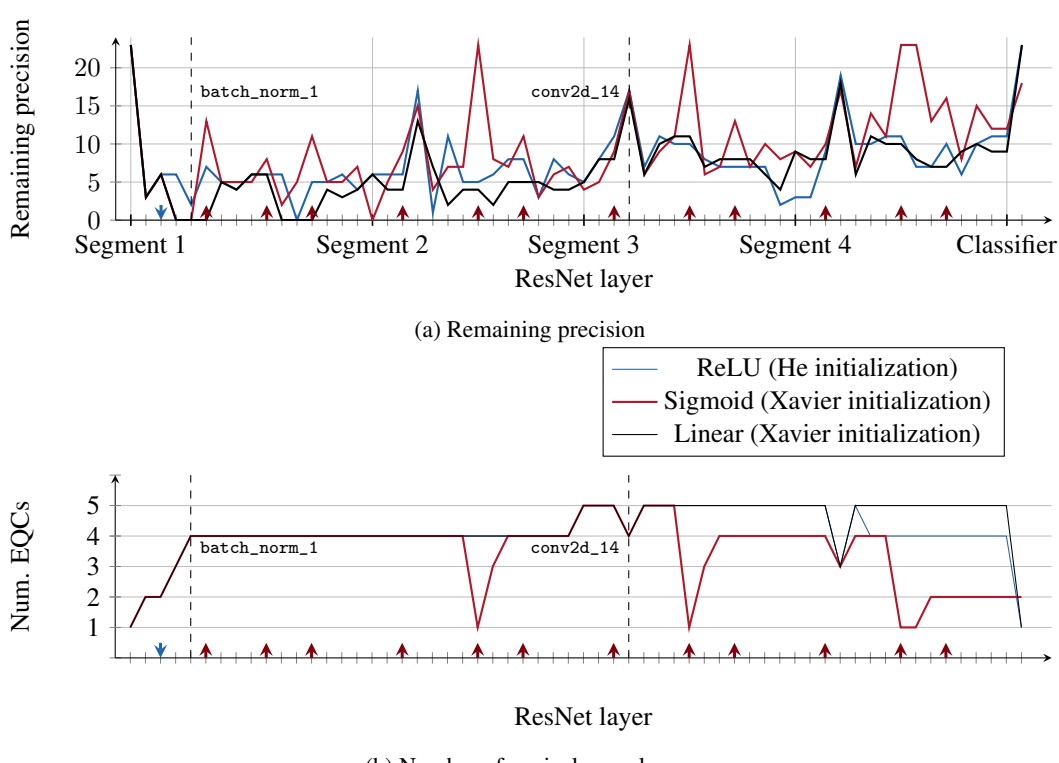

(a) Remaining precision

(b) Number of equivalence classes

Figure SUP-6: Variant of Figure 5 of the main paper, with initialized weights and no training. Sigmoid uses Xavier initialization, ReLU uses He. Linear uses initialized weights of the sigmoid model.

Figure SUP-5 shows the results for model `Cifar10-small`, which features a second pooling layer. The figure is structured the same way as Figure SUP-4. For the `Cifar10-small` model, sigmoid activation reduces the remaining precision in three out of four cases. `MaxPool` increases the remaining precision for both layers, whereas `AvgPool` increases it for one and leaves it unaffected for the other.

Because the reviewer explicitly mentioned activation functions, we also include a variant of the graphic with initialized weights and no training, shown in Figure SUP-6 for the `Cifar10-R18` model, and in Figure SUP-7 for the `Cifar10-small` model. Remaining precision for the `Cifar10-R18` model without training fluctuates across the entire possible range $[0, 23]$. A remaining precision of 23 indicates only a single EQC. The EQC plot in Figure SUP-6b shows that even after

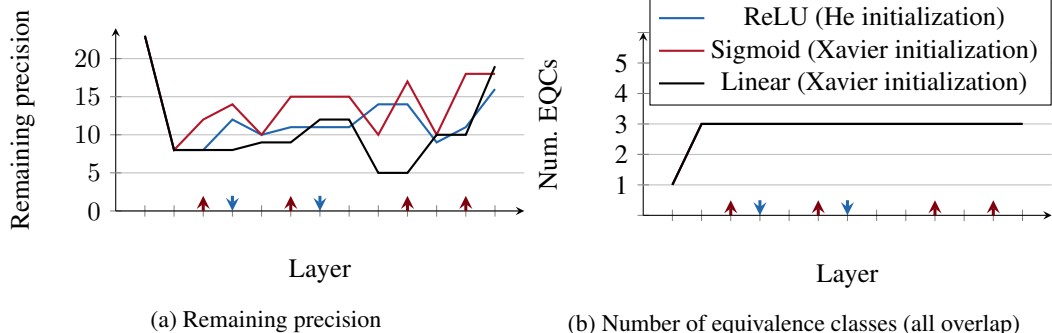

(a) Remaining precision    (b) Number of equivalence classes (all overlap)

Figure SUP-7: Variant of Figure SUP-5 with initialized weights and no training. Methodology is identical to Figure SUP-6.

Table SUP-8: Number of deviations before and after activation layers in `Cifar10-R18` and `Cifar10-small`. Deviations are counted over the flattened output of the layer, and averaged across used in Tables SUP-1 and SUP-2. A value deviates if it is not identical for all hardware platforms.

| Model name | Deviations | | | |
|---|---|---|---|---|
| | ReLU | | Sigmoid | |
| Layer index | Before | After | Before | After |
| `Cifar10-R18` | | | | |
| 5 | 95.3 % | 38.4 % | 94.1 % | 79.5 % |
| 9 | 97.5 % | 34.6 % | 97.7 % | 87.8 % |
| 12 | 98.5 % | 33.8 % | 89.9 % | 80.8 % |
| 18 | 99.3 % | 49.7 % | 99.5 % | 91.7 % |
| 23 | 98.6 % | 43.1 % | 98.4 % | 88.2 % |
| 26 | 97.9 % | 35.2 % | 96.4 % | 87.9 % |
| 32 | 97.9 % | 47.2 % | 98.5 % | 91.8 % |
| 37 | 93.9 % | 43.3 % | 93.1 % | 70.8 % |
| 40 | 75.6 % | 30.8 % | 19.9 % | 7.2 % |
| 46 | 90.8 % | 61.0 % | 96.5 % | 96.0 % |
| 51 | 71.4 % | 54.8 % | 97.9 % | 97.5 % |
| 54 | 79.9 % | 30.1 % | 12.5 % | 8.9 % |
| `Cifar10-small` | | | | |
| 2 | 86.4 % | 49.8 % | 72.3 % | 44.0 % |
| 5 | 97.0 % | 37.6 % | 91.7 % | 71.2 % |
| 9 | 96.6 % | 26.8 % | 82.3 % | 63.8 % |
| 11 | 99.2 % | 30.5 % | 93.2 % | 77.1 % |

all EQCs collapse, new EQCs can arise. This implies that similar behavior is possible for trained weights, and having identical results in intermediate layers does not guarantee identical results in subsequent layers.

Notably, we find more cases where sigmoid activation increases remaining precision for the untrained model. A possible cause for this is the fact that the initialized weights have a lower energy (sum of values), which causes the sigmoid activation to be closer to its linear regime.

In addition to the figures, we count the ratio of deviating values before and after activation. Table SUP-8 shows the results for the trained models.

Table SUP-9: Number of layers that increase, decrease, or leave the remaining precision unaffected. Results are averaged over all samples used in Tables SUP-1 and SUP-2.

| | ReLU | | | | | Sigmoid |
|---|---|---|---|---|---|---|
| Model name | Increase | Unaffected | Decrease | Increase | Unaffected | Decrease |
| Cifar10-R18 | 6.000 | 6.000 | 0.000 | 7.667 | 2.000 | 2.333 |
| Cifar10-small | 3.000 | 1.000 | 0.000 | 1.000 | 0.333 | 2.667 |

**Altering the network's size**

**Request:** "Was any experiment performed to alter the depth/size of the network under test, to see if that would impact the probability of a divergence occurring as depth increased?"

There is no experiment that explicitly cuts, shrinks or enlarges middle layers to investigate the effects. However, the results in Figure 5 were obtained by outputting intermediate layer results, and the results in Figure 4 can be interpreted as varying the size of a single convolutional layer. Both figures show clear trends on how the size affects the number of EQCs as well as the remaining precision. Our analysis of TensorFlow and its underlying Eigen library tells us that actually cutting the layers would result in the same remaining precision and EQCs as shown in Figure 5 because implementation choice on CPUs depends only on the hardware and not on the model. On GPUs, cutting layers will affect these metrics, as early layers will take up memory on the GPU and affect the microbenchmarks of subsequent layers.