# OpenReview forum: "Causes and Effects of Unanticipated Numerical Deviations in Neural Network Inference Frameworks"
_NeurIPS.cc/2023/Conference — NeurIPS 2023 poster_

### Official Review · Reviewer_s6bp · 2023-06-26

**Soundness:** 3 good
**Presentation:** 3 good
**Contribution:** 2 fair
**Rating:** 5
**Confidence:** 4

**Summary:**

The authors a wide range of hardware platforms including both CPUs and GPUs and found inference result (bitwise) of ML model is not consistent across the platforms and even non-deterministic on the same platform. The authors identified the causesof these numerical deviations and attibuted them to accumulation/aggregation rounding errors due to finite precision and different convolution algorithms.

**Strengths:**

* The authors have studied a wide range (75) of hardware platforms, including both CPUs and GPUs, which represents the majorities of ML inference platforms and quatified the deviations through EQCs and Remaining Precisions

* The causes of numerical deviations on different platforms are clearly presented, e.g., CPUs mainly due to precision issues of various parallelisms (SIMD/Multi-cores), and GPUs additionally due to convolution algorithms

**Weaknesses:**

* The numerical deviations author disclosed are well known to the industry, e.g., arithmetic precisions due to accumulation/aggregation order/times and various convolution algorithms. What's new to me is the non-determinism of convs is actually due to runtime variance of microbenchmarks, however which can be surpressed by forcing Frameworks to enable determinism like in TensorFlow (as suggested by authors at line 300) or Pytorch, which is a common practice when serving models. So I don't think authors found significant new sources of numerical deviations.

* Though authors provided a metric "Remaining Precisions" to quantify numerical deviations, it is unclear how it translates to the final inference accuracy/capacity of the model. In the end, unlike traditional software development or high performance computing, ML applications are more robust to numerical deviations in the neurons, and people are even considering "mortal computing" through analog devices. So while "Remaining Precisions" might be insightful to computer arithmetic community, it doesn't directly reflect the capacity of a hardware/software system for ML applications. Studies to bridge the connection between "Remaining Precisions" and accuracies might be desired.

* The takeaway/value of this work is not clear for broader audience of NeurIPS readers. While it reveals the causes of numercial deviations across platforms, and might be valuable to certain domains like security (line 293-296), it doesn't provide a methodology for general audience to benchmark/quantify the numerial precisions of new hardware/softwares, e.g., transformer engines of H100. For example, users could have collected average numerical results of a model from a few mature platforms and run it on a new hardware and compare the remaining precisions, which might refelect how the reliability of a new hardware.

* If my understand is correct, all cases studied are single CPU/GPU and mainly on single floating point precisions, however, in practice, a lot of ML workloads are run in quantized version (bf16/fp16/int8, etc) and in a distributed fashion, which also heavily rely on the ML framework/versions. These might present a first order deviations, compared to the SIMDs/Conv Algos studied by authors.

**Questions:**

The remaining precisions remain relatively stable (don't diminish to 0) throught the entire model (Figure 5) and the authors attribute it to activaiton layers like ReLU/Pooling reduce information and the author also indicated that EQCs are not related to weight distributions(line 273).

Theoretically, if weights are properly distributed (e.g., like Xavier/He initialization that scales to sqrt(accumulation size)) both the activations and their deviations should follow the same distribution (with similar mean/var) throughout the model, regardless of what activation functions are used. So one question is whether the authors have checked if the weight distributions indeed have such property. And to verify the stable remaining precision is indeed due to ReLU/Pooling that reduce information, can the authors try Xavier initialization + sigmoid activation (instead of He + ReLU in typical Resnet) or replace MaxPool to MeanPool to see if this behavior still holds. This should tell which is the cause of non-diminishing remaining precision.

**Limitations:**

The authors have talked about a few in line 315, additionally, as discussed by last point in Weakness, the study is limited to single device and single precision, and these may not be a significant source of deviations compared to distributed systems and quantizations.

---

> ### Author Rebuttal · Authors · 2023-08-09
>
> Thank you very much for your careful analysis of our paper!
>
> > "The numerical deviations author disclosed are well known to the industry [...] So I don't think authors found significant new sources of numerical deviations."
>
> Clearly, the sources of numerical deviations are known. This is why we do not claim to have identified new sources of numerical deviations. Instead, we offer the so-far most comprehensive evaluation of causes and effects of (known) numerical deviations in CNN inference, spanning a wide range of relevant platforms. The novelty lies, among other things, in associating them to properties under control of the machine learning engineer, such as layer type or activation function.
>
> > "[…] What's new to me is the non-determinism of convs is actually due to runtime variance of microbenchmarks, however which can be surpressed by forcing Frameworks to enable determinism […]"
>
> Indeed, some frameworks allow users to enable deterministic operations. This disables microbenchmarks and selects the first supported algorithm instead of the fastest one. The selected algorithm, however, can again vary between GPUs, especially of different generations. Thank you for pointing this out. We will clarify this in the final version of the paper.
>
> > "[…] it is unclear how it ["Remaining Precisions"] translates to the final inference accuracy/capacity of the model. […]  while "Remaining Precisions" might be insightful to computer arithmetic community, it doesn't directly reflect the capacity of a hardware/software system for ML applications. […] "
>
> Our remaining precision metric is not meant for measuring system-level accuracy. The metric is intended to facilitate analysis and discussion of deviations. It is a refinement of the simple (but crude) EQC metric and is much easier to reason about than, e.g., difference norms and cosine distances.
> Moreover, while users are rarely affected *at this point in time*, they might be in the future when ML results are used other applications that require deterministic and reproducible inputs.
>
> > "[…] might be valuable to certain domains like security (line 293-296), it doesn't provide a methodology for general audience to benchmark/quantify the numerial precisions of new hardware/softwares, e.g., transformer engines of H100. […]"
>
> Our understanding is that a broad audience is indeed concerned about security and replicability. By providing our infrastructure on GitHub, we enable other researchers to use our metrics and codebase to trace deviations through the model and produce visualizations like Fig. 5.
> New hardware, such as H100 (released just before the submission deadline, and only very recently available on AWS), can be compared to existing hardware for concrete models. As demonstrated in the paper, deviations have multiple causes at various level of the stack. A single metric that can predict the reliability of new hardware in general may not exist.
>
> > "[…] all cases studied are single CPU/GPU and mainly on single floating point precisions, however, in practice, a lot of ML workloads are run in quantized version (bf16/fp16/int8, etc) and in a distributed fashion […].",
> "[…] the study is limited to single device and single precision, and these may not be a significant source of deviations compared to distributed systems and quantizations."
>
> Thank you for pointing this out. The primary goal of this paper is to answer the question if inference is deterministic and to identify causes of non-determinism. For this reason, we chose the approach of analyzing single CPU/GPU machines. Our measurement and orchestration code enables researchers to analyze multi CPU/GPU machines without much effort.
>
> Please recall that quantization to fp16 is covered in Figure 6 of the paper. As bfloat16 and int8 are not directly compatible with our method of converting the models, we were not able to generate results in time for this rebuttal but will include them in the camera-ready version of the paper.
>
> > "[…] if weights are properly distributed (e.g., like Xavier/He initialization that scales to sqrt(accumulation size)) both the activations and their deviations should follow the same distribution […] So one question is whether the authors have checked if the weight distributions indeed have such property.
> > […] to verify the stable remaining precision is indeed due to ReLU/Pooling that reduce information, can the authors try Xavier initialization + sigmoid activation (instead of He + ReLU in typical Resnet) or replace MaxPool to MeanPool to see if this behavior still holds."
>
> Thank you for this observation. We ran new experiments to confirm that choosing Sigmoid/MeanPool (Xavier initialization) over ReLU/MaxPool (He initialization) does not change the pattern of remaining precision throughout the model.  The results of this experiment are shown in Figures R1 and R2 of the attached PDF. The first and last Sigmoid activation layers decrease the remaining precision, whereas the remaining layers increase or do not affect it. This confirms our reasoning that activation functions simply propagate deviations. The single pooling layer after the first convolution increases the remaining precision for both MaxPool and MeanPool.
> We also measure the effect for a small model with a second pooling layer and find that Sigmoid activation reduces the remaining precision in three out of four cases. MaxPool increases the remaining precision for both layers, whereas MeanPool increases it for one and leaves it unaffected for the other layer.
>
> Moreover, Table R1 shows the influence of activations on the remaining precision and Table R2 compares the number of deviations before and after activations for each layer.
>
> Again, thank you for your valuable input. It helped us to improve the quality and clarity of our work!
>
> If you have any further suggestions or questions, please do not hesitate to let us know.

---

> > ### Comment · Reviewer_s6bp · 2023-08-14
> >
> > Thank you for your detailed explanation, in addition to the experiments you run according to my suggestions. They are highly appreciated.
> >
> > 1. Comments on the new experiment you ran first. Based on the result, my read is there is no significant difference between ReLU/Sigmoid or MaxPool/MeanPool, and you said,
> > > This confirms our reasoning that activation functions simply propagate deviations.
> >
> > if both relu/sigmoid just propagate deviations rather than reducing it, based on your hypothesis in line 206,
> > > convolutional and other parallel data processing layers tend to introduce deviations.
> >
> > then we should expect an monotonic decrease of remaining precision. Otherwise if both relu/sigmoid can reduce deviations, what's the reason behind sigmoid? My take from the experiment is with proper initialization, the remaining precision would stay relatively constant regardless of the activation/pooling you used. Maybe you can just remove activation/pooling completely with Xavior and see how the remaining precision changes. Without a clear understanding of this, the first rebuttal point,
> > > associating them to properties under control of the machine learning engineer, such as layer type or **activation** function.
> >
> > can hardly be justified.
> >
> > 2. Regarding precision. It is true that fp16 data are shown in Figure 6 and related section, however if my undertand is correct, the majority of the study is carried out with fp32 and so are the conclusions. Although the same experiment methodology should be able to be applied to fp16 directly, and potentially similar observations would be found, it should be presented to the readers directly in the paper rather than having readers to carry out experiments themselves. Similar comments apply to a distributed system:
> > > Our measurement and orchestration code enables researchers to analyze multi CPU/GPU machines without much effort.
> >
> > Without these data points I can hardly justify the work has been focusing on the 1st order causes of numerical deviations.
> >
> > 3. Regarding the broad impact and audience. Again, I highly appreciate the tremendous of amount of platforms authors have studied and I believe this can serve as a good tech report to many system engineers. I am just not convinced that the insights it revealed is of great interests to the audience of NeurIPS. I think this is a bit subjective and I will leave to ACs and other reviewers to make the call.

---

> > > ### Author Response · Authors · 2023-08-18
> > >
> > >
> > >
> > > > Thank you for your detailed explanation, in addition to the experiments you run according to my suggestions. They are highly appreciated.
> > >
> > > You are welcome and thank you for your response.
> > >
> > > Concerning our additional experiments:
> > >
> > > > then we should expect an monotonic decrease of remaining precision
> > >
> > > It is not straightforward to reason about the monotonicity of remaining precision (and we don't do this in the paper).
> > > First, for any two EQCs, the remaining precision can increase or decrease by simple addition of a bias.
> > > Here is an example for a hypothetical 4-bit mantissa: 0010 vs 0011 -> RP 3; adding bias 0001 we obtain:  0011 vs 0100 -> RP 1.
> > > Second, remaining precision is a tail metric in Fig. 5, see line 76 in the paper.
> > > Because it is not easy to reason about remaining precision, we included tables R1 and R2 in the rebuttal and would like to bring them to your attention again.
> > > Table R1 shows a difference in the effect on remaining precision between the ReLU and sigmoid activation functions (both, as always, initialized with the correct distribution).
> > > This supports our statements in lines 209-211.
> > > To avoid the difficulty of interpreting aggregate remaining precision, table R2 simply counts the number of deviations.
> > > Observe the difference between ReLU and sigmoid in terms of how many deviations they eliminate.
> > >
> > > > [...] can hardly be justified.
> > >
> > > We hope, with the above explanation, you find the evidence in tables R1 and R2 convincing.
> > >
> > > Regarding the floating point precision, we would like to refer to our rebuttal to Reviewer u1sW:
> > >
> > > > To address your comment, we extended the [fp16 and fp64] experiment to all three models used in the paper.
> > > > We find that the pattern we report holds across all tested models and samples. [...]
> > > > These results will be included in the supplemental material.
> > >
> > > Choosing fp32 as a baseline is still reasonable given that all major platforms use it as default.
> > >
> > > > I believe this can serve as a good tech report to many system engineers.
> > >
> > > Thank you for seeing value in our research.
> > >
> > > > I think this is a bit subjective and I will leave to ACs and other reviewers to make the call.
> > >
> > > For this to happen, we would kindly ask you to consider revising your score from one that suggests technical flaws to a neutral one.

---

> > > > ### Comment · Reviewer_s6bp · 2023-08-18
> > > >
> > > > Thanks for the replies again. Given you observed similar patterns for fp16 and if you could demonstrate it in your final manuscript, I am happy to raise my score to 4. And if you could
> > > >
> > > > > Maybe you can just remove activation/pooling completely with Xavior and see how the remaining precision changes.
> > > >
> > > > I am happy to further increasing my score.

---

> > > > > ### Author Response · Authors · 2023-08-20
> > > > >
> > > > > Thank you for your comment, and for updating your rating.
> > > > >
> > > > > > Maybe you can just remove activation/pooling completely with Xavior and see how the remaining precision changes.
> > > > >
> > > > > To address this request, we changed the activation functions in the sigmoid model to linear.
> > > > > We used the trained weights (originally initialized with Xavier).
> > > > > The pooling layer was left in place to keep the dimensions compatible with the other tested variants.
> > > > >
> > > > > As we cannot update the submitted PDF, please find the updated results [here](https://www.dropbox.com/scl/fi/9ce9y8eepog4ibnulfpky/rebuttal.pdf?rlkey=nr6ew3b3e79dv5tkzyyywjwyy&dl=0).
> > > > >
> > > > > Observe that the remaining precision drops to lower values than for both activation functions, e.g., as low as 1 bit in Figure R1.
> > > > > Across layers, the remaining precision tends to decrease over the course of a ResNet segment, but not monotonically for the reasons elaborated in our previous response.
> > > > > For the smaller model in Figure R2, the difference is less distinct.
> > > > >
> > > > > Note that the final softmax layers of both doctored models produces identical outputs, implying a collapse to a single EQC and remaining precision 23 for fp32.
> > > > > This is caused by the model producing a label with 100% confidence on all hardware configurations.
> > > > > (For brevity, the EQC plot is still excluded, we will add it to the appendix of any camera-ready version.)
> > > > >
> > > > > Concerning the interpretation, we do not see any indication from this set of experiments that would challenge or contradict the statements made in our paper.
> > > > > At the same time, we are cautious to interpret it as strong additional support because ResNet models with post-hoc linearized activation functions are uncommon, and we find them hard to reason about.
> > > > >
> > > > > We would like to thank you once again for pushing us to provide more evidence for our claims, and for pointing us in promising directions.

---

> > > > > > ### Comment · Reviewer_s6bp · 2023-08-20
> > > > > >
> > > > > > > Observe that the remaining precision drops to lower values than for both activation functions
> > > > > >
> > > > > > My interpretation for this is it introduced larger variance (since activations reduces variance) on the inputs to dense(conv) layers and therefore larger variance on the output of dense layers and reduced remaining precision. Therefore
> > > > > > > This confirms our reasoning that activation functions simply propagate deviations.
> > > > > >
> > > > > > > Continuous functions that preserve information (e. g., sigmoid and softmax) can maintain
> > > > > > or potentially amplify deviations. Functions that reduce information, like the rectified linear unit (ReLU), can have a diminishing effect on deviations.
> > > > > >
> > > > > > I would disagree on this, e.g, even sigmoid seems to have the effect of increasing(reducing) remaining precision (variance). My take from the experiments so far is the remaining precision is basically proportional to the variance of the inputs to each layer (so sigmoid/Xavier and relu/He tend to have similar remaining precision), so follows the same mathematical reasonings of weight initialization. Either way, thanks for conducting so many additional experiments in a short time and I believe they provided more evidences for readers to reason about their conclusions, so I would like to raise my final score to 5.

---

> > > > > > > ### Author Response · Authors · 2023-08-21
> > > > > > >
> > > > > > > Thank you again for your response.
> > > > > > >
> > > > > > > For completeness, we have added measurements for initialized weights without training in Figures R3 and R4 in [the linked PDF](https://www.dropbox.com/scl/fi/9ce9y8eepog4ibnulfpky/rebuttal.pdf?rlkey=nr6ew3b3e79dv5tkzyyywjwyy&dl=0).
> > > > > > >
> > > > > > > > Either way, thanks for conducting so many additional experiments in a short time and I believe they provided more evidences for readers to reason about their conclusions, so I would like to raise my final score to 5.
> > > > > > >
> > > > > > > Thank you for your feedback, and for raising your score.
> > > > > > > We are happy to follow up later, if you are interested.

---

### Official Review · Reviewer_3gga · 2023-07-05

**Soundness:** 4 excellent
**Presentation:** 3 good
**Contribution:** 4 excellent
**Rating:** 7
**Confidence:** 5

**Summary:**

This paper explores why the same code & data can result in different results from a trained neural network on multiple different, or even the same, architectures. Considering CPU, GPU, and algorithmic implementation, the paper isolates several key factors that cause variance in the calculated results, which hardware group together, which can cause variation on the same hardware, and eliminate other factors as confounders.

This really is a great paper. I could go into more detail spitting the nuances back at the author, but this is excellent and valuable needed within the ML/DL community.

**Strengths:**

1. Tackles a core issue and question of reproducibility within the research community
2. Performs an extensive analysis requiring immense work to obtain many different hardware platforms to perform the tests.
3. Identified surprising results on the impact of low-precision in mitigating the issue but in unstable ways.


**Weaknesses:**

1. There is some missing related work that should be incorporated for scholastic completeness. In particular the history of numerical differences in ML was documented in "A Siren Song of Open Source Reproducibility" - which covers some older history of related work, but also has two recent works that must also be included: "Problems and Opportunities in Training Deep Learning Software Systems: An Analysis of Variance" and "Randomness In Neural Network Training: Characterizing The Impact of Tooling". Neither work was as thorough in platforms and nailing down what exactly the problem is as this work, so I see this as no barrier to accepting the paper. But we should give completeness to the work by citing the ones who have documented this space previously.
2. The work doesn't fully prepare the reader for their journey in the introduction. It would help the reader to summarize the results in the introduction, and then the last paragraph can explain where/how these results will be reached.

**Questions:**

1. Was any experiment performed to alter the depth/size of the network under test, to see if that would impact the probability of a divergence occurring as depth increased?

**Limitations:**

There are no limitations worth noting.

---

> ### Author Rebuttal · Authors · 2023-08-09
>
> Thank you for pointing us to additional related work, which we unfortunately overlooked when doing our literature search.
>
> Zhuang et al.’s work on training variance gives valuable insights on the training process as a whole, whereas we focus on pinpoint observations concerning inference. Combining their methodology with our instrumentation is promising future work, and we will indicate it as such.
>
> Pham et al. review the influence of implementation-level variance during training, including the same observations that led to our study of microbenchmarks, on the final performance of the trained models. They also survey how often such variances are mentioned and accounted for in the literature, as well as how prevalent knowledge of such variances is among ML researchers. We will cite their paper in our coverage of microbenchmarks.
>
> Raff et al.’s discussion of the influence of code on reproducibility and replicability focuses on the process of ML research. We share the sentiment that current practices of publishing research code are often insufficient. We will include these points in the discussion and cite the reference.
> We will expand our discussion section by the points raised in all three papers to highlight the problem of lacking reproducibility, and in fact the lack of a common definition of this term in the first place.
>
> > "The work doesn't fully prepare the reader for their journey in the introduction. It would help the reader to summarize the results in the introduction, and then the last paragraph can explain where/how these results will be reached."
>
> Thank you for pointing this out. We will restructure the introduction to make it perfectly clear what results will be presented, and in what order.
>
> > "Was any experiment performed to alter the depth/size of the network under test, to see if that would impact the probability of a divergence occurring as depth increased?"
>
> There is no experiment that explicitly cuts, shrinks or enlarges middle layers to investigate the effects. However, the results in Figure 5 were obtained by outputting intermediate layer results, and the results in Figure 4 can be interpreted as varying the size of a single convolutional layer. Both figures show clear trends on how the size affects the number of EQCs as well as the remaining precision.
> Our analysis of TensorFlow and its underlying Eigen library tells us that actually cutting the layers would result in the same remaining precision and EQCs as shown in Figure 5 because implementation choice on CPUs depends only on the hardware and not on the model.
> On GPUs, cutting layers will affect these metrics, as early layers will take up memory on the GPU and affect the microbenchmarks of subsequent layers.
>
> Unfortunately, the page limit of NeurIPS forced us to aggressively select which questions we answer in the paper. We hope to convince you that our focus on breadth (for a comprehensive coverage of causes) while compromising on depth (concerning potential interactions between causes) is the right choice at this stage of the research.
>
> If you have any further suggestions or questions, please do not hesitate to let us know.

---

> > ### Comment · Reviewer_3gga · 2023-08-12
> >
> > >Unfortunately, the page limit of NeurIPS forced us to aggressively select which questions we answer in the paper. We hope to convince you that our focus on breadth (for comprehensive coverage of causes) while compromising on depth (concerning potential interactions between causes) is the right choice at this stage of the research.
> >
> > No convincing was necessary, was just curious. I think the above would be useful for an appendix - I don't think it's conclusive enough to make a hard statement, but its never the less valuable for future work.
> >
> > I think (7) is still the right score for this paper, but whole heartedly encourage my fellow reviewers to raise theirs.

---

### Official Review · Reviewer_u1sW · 2023-07-06

**Soundness:** 3 good
**Presentation:** 3 good
**Contribution:** 3 good
**Rating:** 6
**Confidence:** 3

**Summary:**

The paper explores the causes and effects of the same model and data for inference on different platforms that result in different variations of numerical values. Furthermore, many different hardware specifications within the cloud which are mostly CPU platforms are chosen to evaluate different factors that cause this discrepancy. For the CPU environments, the emphasis is on parallelism and certain Instructions while for the GPU, the focus is on the convolution algorithm that is selected by the GPU at a given time with respect to multiple factors. Finally, this paper offers some mitigation strategies to solve these issues.

**Strengths:**

++ Good effort to separate each factor of deviation for evaluation

++ High diversity of CPUs used for the experiments

++ Most of the graphs are informative and hold useful pieces of information

++ EQC is a good metric to measure diversity

**Weaknesses:**

– The abstract and introduction section could be more organized and well written

– Minor Writing issues(transitions, cohesion, word usage)

– More description of related work could be utilized to differentiate in detail between the paper and other works in the literature

– Real environments might take multiple factors all at once to generate different inference results. This real-world scenario could be explored in the experiments instead of evaluating each factor separately

**Questions:**

C1. It would be better to list the contribution of the work in the introduction. Also since this paper is an analysis of cause and effect, it would be better to evaluate some of the mitigation strategies to show effectiveness.

C2. Although the title suggests a general analysis of neural networks, most of the analysis is about layers of CNN or ResNet and ways of calculating convolution. Therefore it would be better to either analyze other neural networks as well or indicate this in the introductory sections of the paper.

C3. It would be better to have a list of CPU flags and corresponding clusters as an appendix. Also, More description on how often certain SIMID flags within a cluster are activated would help the readers to understand the importance of SIMID instructions for differences in inference results.

C4. Figure 6 might need more explanation regarding the importance of floating point precision. Also Maybe other Neural Networks could be tested to see if they follow the same pattern for single-point precision.

C5. More description about how different the convolution algorithms do the calculations might extract new insights into how they affect EQCs. Also, some results might confuse the reader, for example, how different explicit and implicit GEMMs are and the reasoning behind a GPU choosing one over the other.

**Limitations:**

See Questions.

---

> ### Author Rebuttal · Authors · 2023-08-09
>
> Thank you for your remarks regarding the organization of sections, writing issues, and description of related work. We will update the revised version of the paper to ensure the contributions of our research are well-articulated.
>
> > "Real environments might take multiple factors all at once to generate different inference results. This real-world scenario could be explored in the experiments instead of evaluating each factor separately."
>
> In addition to our study of isolated influences, our main results in Figure 1 of the paper, as well as Tables 1 and 2 in the supplemental material, in fact present the results of a large-scale validation study using real models in real CPU and GPU environments. We will revise line 92 to explicitly state that these measurements capture the interplay of all influences in real-world scenarios.
>
> > C1. It would be better to list the contribution of the work in the introduction. Also since this paper is an analysis of cause and effect, it would be better to evaluate some of the mitigation strategies to show effectiveness.
>
> We will include a clear and concise description of the main findings in the introduction. We will also highlight that our results of Section 3.4 do evaluate a mitigation strategy (varying the floating point precision) and analyze its effectiveness.
>
> > C2. Although the title suggests a general analysis of neural networks, most of the analysis is about layers of CNN or ResNet and ways of calculating convolution. Therefore it would be better to either analyze other neural networks as well or indicate this in the introductory sections of the paper.
>
> This assessment is fair. We will clarify the scope of our work in the introduction. By publishing the measurement and orchestration code we make it easy for researchers to extend our research to other layer types and models. As an indication that our findings generalize, our experiments for Figure 5 show that batch normalization can introduce deviations.
>
> > C3. It would be better to have a list of CPU flags and corresponding clusters as an appendix.
>
> We have generated a full table including all CPU flags and added it to the supplemental material. (Unfortunately, we cannot provide it here as it would exceed the page limit for the rebuttal document.)
>
> > C3. (cont.) Also, More description on how often certain SIMID flags within a cluster are activated would help the readers to understand the importance of SIMID instructions for differences in inference results.
>
> We fully agree with your comment and have in fact made efforts to measure the usage of SIMD instructions in TensorFlow. However, despite numerous attempts at dynamic instrumentation (gdb, rr, valgrind/cachegrind, and TensorFlow's own profiler) and code analysis (of TensorFlow and its underlying Eigen library), we were unable to obtain reliable results due to the complexity of TensorFlow's monolithic architecture. While certainly desirable, we believe that the value of our work does not hinge on this description.
>
> > C4. Figure 6 might need more explanation regarding the importance of floating point precision. Also Maybe other Neural Networks could be tested to see if they follow the same pattern for single-point precision.
>
> We recognize the need for more explanation of Figure 6 and will provide a more detailed description in the final version of the paper. To address your comment, we extended the experiment to all three models used in the paper. We find that the pattern we report holds across all tested models and samples. Due to the 1-page limit of the rebuttal document we cannot include dendrograms, but can confirm that the EQCs follow the same patterns as shown in Figure 6. These results will be included in the supplemental material.
>
> > C5. More description about how different the convolution algorithms do the calculations might extract new insights into how they affect EQCs.
>
> Due to the page limit we omitted a detailed description of calculations of different convolution algorithms. We do agree that they might provide insights into their effect on EQCs and have therefore included them in the supplementary material along with references to the relevant literature and vendor documentation.
>
> > C5. (cont.) Also, some results might confuse the reader, for example, how different explicit and implicit GEMMs are and the reasoning behind a GPU choosing one over the other.
>
> It seems that our description of the difference between explicit and implicit GEMM in line 153 has been imprecise (specifically "presumably"). Thank you for the comment, we will refer to the literature and clarify that the only difference is that explicit GEMM stores the Toeplitz matrix in memory, whereas implicit GEMM computes it on the fly.
>
> Thank you for bringing these points to our attention.
> If you have any further suggestions or questions, please do not hesitate to let us know.

---

> > ### Comment · Reviewer_u1sW · 2023-08-19
> >
> > Thank you for the detailed response. Many of my concerns have been addressed. I have raised my rating from 4 to 6.

---

### Official Review · Reviewer_djPp · 2023-07-06

**Soundness:** 3 good
**Presentation:** 3 good
**Contribution:** 2 fair
**Rating:** 5
**Confidence:** 4

**Summary:**

The paper presents an extensive empirical study on the numerical instabilities for convolutions across different platforms. The findings of why these instabilities occur are interesting and informative. They further cluster them into equivalence classes for ease of explanation.

**Strengths:**

Extensive evaluation on a large number of platforms across both CPUs and GPUs.

**Weaknesses:**

The results and findings are very interesting, but the utility values of the EQCs and the errors they find is not demonstrated. Essentially, how do you use this information to make inference more robust for example?

No results to corroborate line 285. Will these imperfections lead to different model outputs? Some of these errors are tolerable specially in classification setting. Authors themselves mention this is rare. Some evidence or experiment on flips would be useful.

**Questions:**

I find the paper very interesting to read. It is not a research solutions paper, but an extensive study on numerical stabilities. Even though, the research contribution on novel techniques do not exist, I find the community would benefit from these findings the paper reports through extensive experimentation.

Please answer the following questions in the rebuttal.

* Why is Toeplitz based convolution lossy?
* How significant are the loss of precision? Does it affect results in inference? Can this be trained away by fine tuning?
* How do you use the EQCs to better mitigate any inaccuracies in inference?
* Algorithmic choice in GPUs are anyway supposed to get different results, since they are different approximations of the convolution. I find this expected and obvious.


**Limitations:**

Authors mention the limitations explicitly under discussion.

---

> ### Author Rebuttal · Authors · 2023-08-09
>
> Thank you for carefully reviewing our paper.
>
> > "The results and findings are very interesting, but the utility values of the EQCs and the errors they find is not demonstrated. Essentially, how do you use this information to make inference more robust for example?"
>
> The concrete utility of this work lies in identifying the root causes of deviations as a necessary first step towards reproducibility.  Our findings give researchers and ML framework developers concrete starting points for improving robustness. Thank you for asking this question and bringing this lack of framing to our attention. We will clarify our contributions in the introduction.
>
> > "No results to corroborate line 285. Will these imperfections lead to different model outputs? Some of these errors are tolerable specially in classification setting. Authors themselves mention this is rare. Some evidence or experiment on flips would be useful."
>
> We agree that our formulation in line 285 is too general, and will revise it to state that bit-level reproducibility of inference results is a necessary requirement for reproducibility. While our study focuses on inference for comprehensibility, the causes we identify may also explain phenomena observed (but not explained) during training, as shown for instance by [Zhuang et al. "Randomness in neural network training: Characterizing the impact of tooling" (2022) and Pham et al. "Problems and opportunities in training deep learning software systems: An analysis of variance." (2020)].
> Experiments on forcing label flips have been described in related work which we cite in line 288.
>
> > "Why is Toeplitz based convolution lossy?"
>
> Our presentation of convolution approaches was imprecise. In fact, convolution with Toeplitz matrices is not lossy. The "lossy" transformations in lines 60 and 63 refer to Winograd and FFT-based convolution, respectively. Thank you for pointing this out, we will revise accordingly.
>
>  > "How significant are the loss of precision? Does it affect results in inference? Can this be trained away by fine tuning?"
>
> As stated in line 287, reductions in common model performance characteristics are unlikely.
> While incorporating deviations into the loss function to reduce deviations by fine-tuning is conceivable in principle, we deem it impractical with current frameworks. The deviations happen at such a low level that the ML framework does not capture them without the kind of instrumentation we built for the purpose of this research. We thank you for this question and will keep it in mind for future work on this topic.
>
> > "How do you use the EQCs to better mitigate any inaccuracies in inference?"
>
> We introduce EQCs - a metric linked to the effect numerical deviations - as a tool to understand their causes. Engineers striving for deterministic and consistent results should aim for a single EQC, by restricting the inference platforms to those within the same EQC or by applying mitigation measures that reduce the number of EQCs. Researchers can use EQCs to test new mitigation measures. Thank you for this question, we will elaborate more on the practical use of EQCs in the paper.
>
> > "Algorithmic choice in GPUs are anyway supposed to get different results, since they are different approximations of the convolution. I find this expected and obvious."
>
> It is known that algorithm choice can vary on GPUs, and that different algorithms yield different results. However, to the best of our knowledge, we are the first to report the variations in chosen algorithms, given the same models, inputs, and hardware.
> As mentioned in line 300, even with "enforced" determinism (through setting determinism-flags in TensorFlow and PyTorch), different algorithms can be chosen based on the algorithms supported by the hardware.
>
> Thank you again for your thorough review and your valuable remarks.
> If you have any further suggestions or questions, please do not hesitate to let us know.

---

> > ### Comment · Reviewer_djPp · 2023-08-18
> >
> > Thank you for addressing my concerns. The authors agree to provide more discussion on the utility of EQCs. That said, I would have liked to see more concrete reasons or experiments to corroborate their claims (e.g., While incorporating deviations into the loss function to reduce deviations by fine-tuning is conceivable in principle, we deem it impractical with current frameworks). I would like to keep my score.

---

> > > ### Author Response · Authors · 2023-08-21
> > >
> > > Thank you for your comment.
> > >
> > > > I would have liked to see more concrete reasons or experiments to corroborate their claims (e.g., While incorporating deviations into the loss function to reduce deviations by fine-tuning is conceivable in principle, we deem it impractical with current frameworks).
> > >
> > > Of course we respect your judgement.
> > > Perhaps we can take this opportunity to elaborate on why fine-tuning for consistency is beyond current technology and would require much more research.
> > > (In hindsight, calling it "impractical" was probably too brief.)
> > >
> > > Recall that most deviations we study can only be observed between different platforms, i.e., different physical machines.
> > > While it is common to measure outside information and incorporate it into the loss function, this is usually done for information that is available on the same machine.
> > > Capturing deviations from different machines is feasible, and our experiments provide the infrastructure to do so.
> > > However, it is not clear how to combine them into a single loss function, and different strategies would have to be evaluated.
> > >
> > > Even if a method for calculating "deviation loss" was found, it might result in different weight updates on different platforms.
> > > This is because the same factors that cause deviations during inference are also present during back-propagation.
> > > (We have identified this issue during followup research in a similar direction as your suggestion.)
> > > Moreover, it is far from certain that fine-tuning against deviations on $k$ platforms - if possible at all for $k>>2$ - would generalize to the $k+1$th platform.
> > >
> > > Another approach would be to come up with a model of the arithmetic processes that cause deviations in order to simulate their occurrence on a single platform.
> > > While such a model would clearly be nice to have, it appears to us that it would require significant new research (and substantial space to document it), including reverse-engineering of closed source GPU software.
> > > We deem that this would go far beyond the scope of our paper.
> > >
> > > In summary, we agree that exploring fine-tuning as a mitigation measure is an interesting idea, and see it as a promising direction for future research motivated by our present work.
> > > We are happy to mention and - space permitting - briefly discuss this the camera-ready version.
> > >
> > > Thank you once again for your review and in particular this suggestion.

---

### Author Rebuttal · Authors · 2023-08-09

We thank all four reviewers for their insightful feedback and thoughtful suggestions. We are very glad that the suggestions by different reviewers are compatible (and partly overlapping). This enables us to incorporate all of them into the final version of the paper.

Responses to specific questions and remarks are posted under the respective reviews.
The attached PDF document contains the figures and tables of the experiments we ran to address specific questions in the reviews.

---

> ### Author Response · Authors · 2023-08-21
>
> We would like to thank all four reviewers again for their continued efforts, insightful comments, and thorough reviews. This has been a very pleasant rebuttal/discussion process.
>
> Kind regards,
> the authors

---

### Decision · Program_Chairs · 2023-09-21

**Decision:**

Accept (poster)

**Comment:**

Reviewers found this paper exciting and unanimously recommended accept.